# The antibiotic bedaquiline activates host macrophage innate immune resistance to bacterial infection

**Alexandre Giraud-Gatineau[1,2], Juan Manuel Coya[3†], Alexandra Maure[1,2†], Anne Biton[4†], Michael Thomson[5], Elliott M Bernard[6], Jade Marrec[3], Maximiliano G Gutierrez[6], Gérald Larrouy-Maumus[5], Roland Brosch[1‡], Brigitte Gicquel[3,7‡], Ludovic Tailleux[1,3]***

[1]Unit for Integrated Mycobacterial Pathogenomics, CNRS UMR 3525, Institut Pasteur, Paris, France; [2]Université Paris Diderot, Sorbonne Paris Cité, Cellule Pasteur, Paris, France; [3]Mycobacterial Genetics Unit, Institut Pasteur, Paris, France; [4]Bioinformatics and Biostatistics, Department of Computational Biology, USR 3756 CNRS, Institut Pasteur, Paris, France; [5]MRC Centre for Molecular Bacteriology and Infection, Department of Life Sciences, Faculty of Natural Sciences, Imperial College London, London, United Kingdom; [6]Host-Pathogen Interactions in Tuberculosis Laboratory, The Francis Crick Institute, London, United Kingdom; [7]Department of Tuberculosis Control and Prevention, Shenzhen Nanshan Center for Chronic Disease Control, Shenzhen, China

**\*For correspondence:**
ludovic.tailleux@pasteur.fr

[†]These authors also contributed equally to this work
[‡]These authors also contributed equally to this work

**Competing interests:** The authors declare that no competing interests exist.

**Abstract** Antibiotics are widely used in the treatment of bacterial infections. Although known for their microbicidal activity, antibiotics may also interfere with the host's immune system. Here, we analyzed the effects of bedaquiline (BDQ), an inhibitor of the mycobacterial ATP synthase, on human macrophages. Genome-wide gene expression analysis revealed that BDQ reprogramed cells into potent bactericidal phagocytes. We found that 579 and 1,495 genes were respectively differentially expressed in naive- and *M. tuberculosis*-infected macrophages incubated with the drug, with an over-representation of lysosome-associated genes. BDQ treatment triggered a variety of antimicrobial defense mechanisms, including phagosome-lysosome fusion, and autophagy. These effects were associated with activation of transcription factor EB, involved in the transcription of lysosomal genes, resulting in enhanced intracellular killing of different bacterial species that were naturally insensitive to BDQ. Thus, BDQ could be used as a host-directed therapy against a wide range of bacterial infections.

## Introduction

Antibiotics are commonly used in the treatment of bacterial infections, and, in effectively combating such diseases, have substantially increased human life expectancy. As with most drugs, antibiotic treatment can also alter host metabolism, leading to adverse side-effects, including nausea, hepato-toxicity, skin reactions, and gastrointestinal and neurological disorders. Such side-effects can become critical when antibiotic treatment is long and involves several drugs, as in the treatment of tuberculosis (TB), where 2–28% of patients develop mild liver injury during treatment with first-line drugs (*Agal et al., 2005*).

Antibiotics can interfere with the immune system, indirectly through the disturbance of the body's microbiota (*Ubeda and Pamer, 2012*), or directly by modulating the functions of immune cells. Such interactions may impact treatment efficacy or the susceptibility of the host to concomitant infection.

**eLife digest** The discovery of antibiotic drugs, which treat diseases caused by bacteria, has been a hugely valuable advance in modern medicine. They work by targeting specific cellular processes in bacteria, ultimately stopping them from multiplying or killing them outright. Antibiotics sometimes also affect their human hosts and can cause side-effects, such as gut problems or skin reactions.

Recent evidence suggests that antibiotics also have an impact on the human immune system. This may happen either indirectly, by affecting 'friendly' bacteria normally present in the body, or through direct effects on immune cells. In turn, this could change the effectiveness of drug treatments. For example, if an antibiotic weakens immune cells, the body could have difficulty fighting off the existing infection – or become more vulnerable to new ones.

However, even though new drugs are being introduced to combat the worldwide rise of antibiotic-resistant bacteria, their effects on immunity are still not well understood. For example, bedaquiline is an antibiotic recently developed to treat tuberculosis infections that are resistant to several drugs. Giraud-Gatineau et al. wanted to determine if bedaquiline altered the human immune response to bacterial infection independently from its direct anti-microbial effects.

Macrophages engulf foreign particles like bacteria and break them down using enzymes stored within small internal compartments, or 'lysosomes'. Initial experiments using human macrophages, grown both with and without bedaquiline, showed that the drug did not harm the cells and that they grew normally. A combination of microscope imaging and genetic analysis revealed that exposure to bedaquiline not only increased the number of lysosomes within macrophage cells, but also the activity of genes and proteins that increase lysosomes' ability to break down foreign particles.

These results suggested that bedaquiline treatment might make macrophages better at fighting infection, even if the drug itself had no direct effect on bacterial cells. Further studies, where macrophages were first treated with bedaquiline and then exposed to different types of bacteria known to be resistant to the drug, confirmed this hypothesis: in every case, the treated macrophages became efficient bacterial killers. In contrast, older anti-tuberculosis drugs did not have any such potentiating effect on the macrophages.

This work sheds new light on our how antibiotic drugs can interact with the cells of the human immune system, and can sometimes even boost our innate defences. Such immune-boosting effects could one day be exploited to make more effective treatments against bacterial infections.

For example, after treatment completion, TB patients are more vulnerable to reactivation and reinfection of the disease, suggesting therapy-related immune impairment (*Cox et al., 2008*). Drug-sensitive TB can be cured by combining up to four antibiotics in a 6-month treatment; specifically, isoniazid (INH), rifampicin (RIF), ethambutol (EMB) and pyrazinamide (PZA) for 2 months, and INH and RIF for additional 4 months. INH induces apoptosis of activated $CD4^+$ T cells in *Mycobacterium tuberculosis* (MTB)-infected mice (*Tousif et al., 2014*) and leads to a decrease in Th1 cytokine production in household contacts with latent TB under preventive INH therapy (*Biraro et al., 2015*). RIF has immunomodulatory properties and acts as a mild immunosuppressive agent in psoriasis (*Tsankov and Grozdev, 2011*). RIF reduces inflammation by inhibiting IκBα degradation, mitogen-activated protein kinase (MAPK) phosphorylation (*Bi et al., 2011*), and Toll-like receptor 4 signaling (*Wang et al., 2013*). PZA treatment of MTB-infected human monocytes and mice significantly reduces the release of pro-inflammatory cytokines and chemokines (*Manca et al., 2013*). Recently, Puyskens et al. showed that several anti-TB drugs bind to the aryl hydrocarbon receptor and may impact host defense (*Puyskens et al., 2020*). It is therefore necessary to understand how antibiotic treatment modulates macrophage functions, and more generally, how it impacts the host immune response.

The worldwide rise in antibiotic resistance is a major threat to global health care. A growing number of bacterial infections, such as pneumonia, salmonellosis, and TB, are becoming harder to treat as the antibiotics used to treat them become less effective. While new antibiotics are being developed and brought to the clinic, their effects on the human immune system are not being studied indepth. Here, we have investigated the impact of a recently approved anti-TB drug, bedaquiline

(BDQ), on the transcriptional responses of human macrophages infected with MTB. Macrophages are the primary cell target of MTB, which has evolved several strategies to survive and multiply inside the macrophage phagosome, including prevention of phagosome acidification (*Sturgill-Koszycki et al., 1994*), inhibition of phagolysosomal fusion (*Armstrong and Hart, 1975*) and phagosomal rupture (*Simeone et al., 2012*; *van der Wel et al., 2007*). They play a central role in the host response to TB pathogenesis, by orchestrating the formation of granulomas, presenting mycobacterial antigens to T cells, and killing the bacillus upon IFN-γ activation (*Cambier et al., 2014*). BDQ is a diarylquinoline that specifically inhibits a subunit of the bacterial adenosine triphosphate (ATP) synthase, decreasing intracellular ATP levels (*Andries et al., 2005*; *Koul et al., 2007*). It has 20,000 times less affinity for human ATP synthase (*Haagsma et al., 2009*). The most common side effects of BDQ are nausea, joint and chest pain, headache, and arrhythmias (*Diacon et al., 2012*; *TMC207-C208 Study Group et al., 2014*). However, possible interactions between BDQ and the host immune response have not been studied in detail. Understanding the impact of BDQ on the host immune response may help to develop strategies aiming at improving drug efficacy and limiting side-effects, including cytotoxicity, alteration of cell metabolism, and immunomodulation.

## Results

### BDQ modulates the response of naïve and MTB-infected macrophages

We treated human monocyte-derived macrophages from four healthy donors with BDQ at 5 µg/mL, which corresponds to the concentration detected in the plasma of TB patients treated with BDQ (*Andries et al., 2005*). This concentration did not affect cell viability over an incubation period of 7 days (*Figure 1—figure supplement 1*). After 18 hr of treatment, we characterized the genome-wide gene expression profiles of BDQ-treated macrophages by RNAseq, with DMSO-treated cells serving as a control. The expression of 579 genes was affected by BDQ (FDR < 0.05, *Figure 1—source datas 1* and *2*), with 186 being upregulated and 393 being downregulated. We classified all 579 genes by performing gene-set enrichment analysis using ClueGO cluster analysis (*Bindea et al., 2009*). The gene set upregulated by BDQ was significantly enriched for genes associated with lysosome, phagocytic vesicle membrane, vacuolar lumen, hydrolase activity and lipid homeostasis (*Figure 1A*).

We next evaluated if BDQ could modulate gene expression in MTB-infected cells. In order to exclude potential differences due to the MTB bacillary load between treated and untreated cells, we generated a virulent BDQ-resistant strain of *M. tuberculosis* (BDQr-MTB). The selected clone, which carried a Ala63→Pro mutation in subunit c of the ATP synthase (*Andries et al., 2005*; *Figure 1—figure supplement 2A*), had a similar generation time to wild-type bacteria when cultured in 7H9 liquid medium, although we observed a slower growth after 7 days of treatment. (*Figure 1—figure supplement 2B*). We also noted no difference in virulence or in intracellular growth of both wild-type and BDQ-resistant MTB (*Figure 1—figure supplement 2C-E*). As expected, the MIC$_{99}$ (defined as the concentration required to prevent 99% growth) for susceptible MTB was 0.07 µg/mL, a value similar to previously published study (*Andries et al., 2005*), while the MIC$_{99}$ of the BDQr-MTB was 36 µg/mL. We infected macrophages with BDQr-MTB. After 24 hr of infection, cells were incubated for an additional 18 hr with BDQ (5 µg/mL). The bacillary load of resistant MTB inside macrophages was the same in untreated cells as in cells after 18 hr of BDQ treatment. In contrast, in the same experiment using BDQ-susceptible MTB, there was a 70% decrease in the bacillary load (*Figure 1—figure supplement 2C*).

Following treatment, we characterized the genome-wide gene expression profiles of MTB-infected macrophages, as described above. The expression of 1,495 genes was affected by BDQ (FDR < 0.05, *Figure 1A*, *Figure 1—source datas 3* and *4*), with 499 being upregulated and 996 being downregulated. More genes were thus affected by BDQ treatment in MTB-infected cells than in naive macrophages. This probably reflects the fact that MTB infection induces an extensive remodeling of the transcriptome (*Barreiro et al., 2012*; *Tailleux et al., 2008*). The genes differentially expressed by BDQ only in MTB-infected macrophages are enriched in genes related to assembly of the endoplasmic reticulum-Golgi intermediate compartment, membrane raft and cellular protein metabolic process (*Table 1*, *Figure 1—source datas 5* and *6*). This probably reflects the cell adaptation to infection. Functional annotation of the gene set upregulated by BDQ also revealed

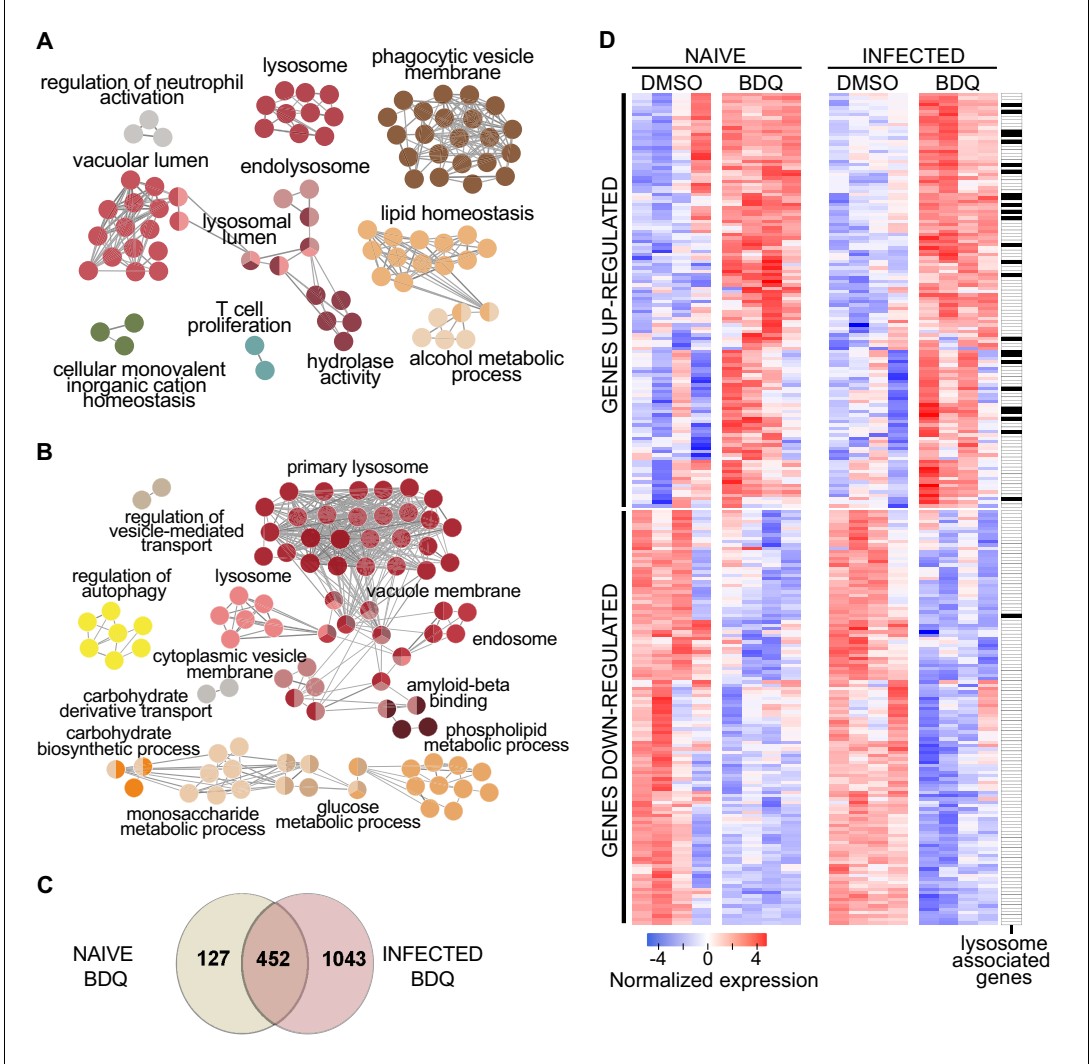

**Figure 1.** BDQ modulates the response of human macrophages. Cells from four individual donors were treated with BDQ (5 μg/mL) for 18 hr. Differentially expressed genes were identified by mRNAseq. (**A**) Gene ontology enrichment analysis of genes whose expression is upregulated by BDQ treatment, using the Cytoscape app ClueGO (FDR < 0.05; LogFC >0.5). (**B**) Cells were infected with BDQ-resistant MTB for 24 hr and then treated with BDQ (5 μg/mL) for an additional 18 hr. Gene ontology enrichment analysis of genes whose expression is up-regulated by BDQ treatment in BDQr-MTB-infected cells, using the Cytoscape app ClueGO (FDR < 0.05; LogFC >0.5). (**C**) Venn diagram showing the number of genes regulated by BDQ treatment in naive and BDQr-MTB-infected macrophages, relative to untreated controls. (**D**) Heatmap showing differential expression of genes differentially expressed by BDQ in naive and BDQr-MTB-infected cells. Each column corresponds to one donor. Data were normalized to determine the log ratio with respect to the median expression of each gene.

The online version of this article includes the following source data and figure supplement(s) for figure 1:

**Source data 1.** Genes whose expression is upregulated in naive macrophages upon BDQ treatment.
**Source data 2.** Genes whose expression is downregulated in naive macrophages upon BDQ treatment.
**Source data 3.** Genes whose expression is upregulated in BDQr-MTB-infected macrophages upon BDQ treatment.
**Source data 4.** Genes whose expression is downregulated in BDQr-MTB-infected macrophages upon BDQ treatment.
**Source data 5.** Genes differentially expressed in BDQr-MTB infected macrophages by BDQ.
**Source data 6.** Genes differentially expressed in naive macrophages by BDQ.
**Source data 7.** Differentially expressed genes both in naive and in BDQr-MTB infected macrophages upon BDQ treatment.
**Figure supplement 1.** Cell viability assay of macrophages incubated with BDQ.
**Figure supplement 2.** Generation of BDQ resistant MTB strain (BDQr-MTB) and evaluation of its virulence.

**Table 1.** Gene Ontology (GO) functional annotation of genes differentially expressed by BDQ only in naïve- and BDQr-MTB-infected macrophages.

**Specific NAIVE BDQ genes**

| GO category | avg. LogFC | p-value |
|---|---|---|
| Cell division | -0.51 | 8.34E-05 |
| Sphingolipid metabolic process | 0.33 | 1.42E-04 |
| Angiogenesis | 0.67 | 5.16E-04 |
| Spindle | -0.63 | 5.46E-04 |
| Lysosomal lumen | 0.35 | 1.21E-04 |
| Glycosphingolipid metabolic process | 0.35 | 1.21E-03 |
| Response to oxidative stress | 0.49 | 1.26E-03 |
| Mitotic cell cycle | -0.58 | 1.29E-03 |

**Specific INFECTED BDQ genes**

| GO category | avg. LogFC | p-value |
|---|---|---|
| Endoplasmatic reticulum-Golgi intermediate compartment | -0.38 | 2.90E-07 |
| Membrane raft | -0.34 | 4.93E-05 |
| Cellular protein metabolic process | -0.35 | 3.32E-04 |
| Lipid binding | -0.36 | 4.88E-04 |
| Ribonucleoprotein complex binding | -0.37 | 5.17E-04 |
| Protein dephosphorylation | -0.34 | 5.43E-04 |
| Lysosomal membrane | -0.34 | 6.31E-04 |
| Ubiquitin-dependent protein catabolic process | 0.74 | 6.48E-04 |

that similar pathways were affected by BDQ in naive and BDQr-MTB-infected macrophages, with an enrichment for genes associated with glucose/phospholipid metabolism and lysosome (*Figure 1B*, *Figure 1—source datas 3* and *4*). 452 genes were differentially expressed in both naive and MTB-infected cells upon BDQ treatment with an over-representation of lysosome-associated genes (*Figure 1C–D*, *Figure 1—source data 7*).

## BDQ affects host metabolism

As metabolic pathways were over-represented in our RNAseq analysis, we investigated if glycolysis is affected by BDQ treatment using the Seahorse Extracellular Flux analyzer. This assay measures the rate of proton accumulation in the extracellular medium during glycolysis (glycoPER) and can discriminate between basal glycolysis, induced glycolytic capacity (by addition of rotenone/antimycin A (Rot/AA), an inhibitor of the mitochondrial electron transport chain), and non-glycolytic acidification (by addition of the glycolytic inhibitor 2-deoxy-D-glycose (2-DG)). After incubation with BDQ, we observed a 30% decrease in basal glycolysis and glycolytic capacity compared to untreated cells (*Figure 2A–B*, *Figure 2—figure supplement 1A–B*).

We assessed phospholipid metabolism, a pathway also identified in our ClueGO cluster analysis (*Figure 1B*). Like glycolysis, lipid metabolism affects macrophage phenotype and function (*Remmerie and Scott, 2018*). We analyzed the lipid profile of BDQ-treated cells using MALDI-TOF mass spectrometry. We observed an increase of phosphatidylinositols upon incubation with BDQ (*Figure 2C*, *Figure 2—figure supplement 1C*). No significant changes were observed in the levels of phosphatidylethanolamines, phosphatidylglycerols, or cardiolipins. Taken together, these data show that BDQ induced a significant metabolic reprogramming of both MTB-infected and resting macrophages.

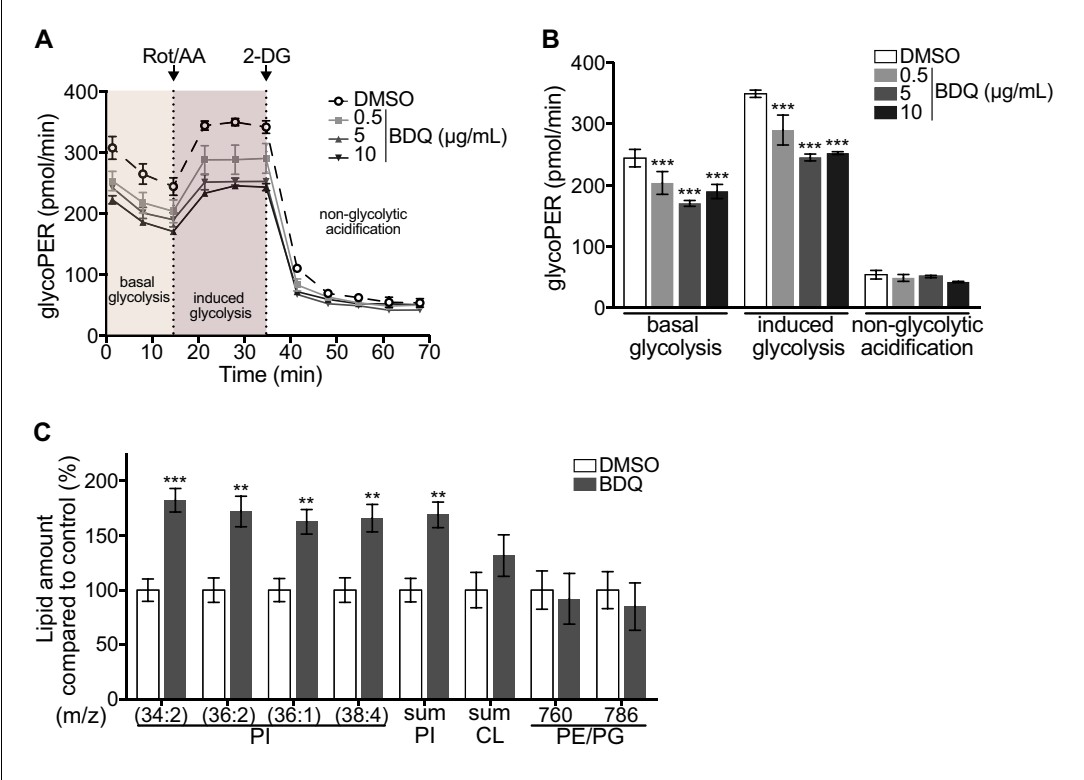

**Figure 2.** Modulation of host metabolism by BDQ. (A–B) The Glycolytic Rate Assay was performed in heat killed-MTB stimulated macrophages treated with BDQ, in the presence of rotenone/antimycin A (Rot/AA) and 2-deoxy-D-glycose (2-DG), inhibitors of the mitochondrial electron transport chain and glycolysis, respectively (one-way ANOVA test). One representative experiment (of two) is shown. (C) Lipid profile of BDQr-MTB infected cells treated with BDQ, by MALDI-TOF (unpaired two tailed Student's t test). PI: Phosphotidylinositol; CL: Cardiolipids; PE: Phosphatidylethanolamine; PG: Phosphatidylglycerol. Numbers correspond to mass-to-charge ratio (m/z). Cells derived from three donors were analyzed. Error bars represent the mean ± SD and significant differences between treatments are indicated by an asterisk, in which *p<0.05, **p<0.01, ***p<0.001.
The online version of this article includes the following figure supplement(s) for figure 2:

**Figure supplement 1.** BDQ modulates the metabolism of naïve macrophages.

## BDQ increases macrophage lysosomal activity

Macrophages are involved in innate immunity and tissue homeostasis through their detection and elimination of microbes, debris, and dead cells, which occurs in lysosomes (*Wynn et al., 2013*). Lysosomes are acidic and hydrolytic organelles responsible for the digestion of macromolecules. Recent work has shown that they are also signaling platforms, which respond to nutrient and cellular stress (*Lawrence and Zoncu, 2019*). Functional annotations based on the GO database of the differentially expressed genes suggested a substantial impact of BDQ treatment on lysosome function (*Figure 1A–B*). We identified 38 and 54 differentially expressed genes by BDQ, respectively in naïve- and BDQr-MTB infected cells (FDR < 0.05, *Figure 3A*). These genes are involved in lysosome biogenesis, transport and degradation of small molecules, and lysosomal acidification. They included genes coding for components of vacuolar ATPase (V-ATPase), hydrolases, and SLC11A1 (NRAMP1), a divalent transition metal transporter involved in host resistance to pathogens, including MTB (*Meilang et al., 2012*).

To validate our transcriptomic data, we incubated BDQ-treated, BDQr-MTB-infected cells with LysoTracker Red DND-99, a red fluorescent probe that labels acidic organelles, and analyzed them using flow cytometry. No differences were observed between control and treatment after 3 hr of BDQ treatment (*Figure 3B*). However, at 18 hr and 48 hr post-treatment, fluorescence intensity was substantially increased in macrophages incubated with BDQ compared to DMSO-treated cells (1.7 and 5.4 times more, respectively). These results were supported by confocal microscopy, which

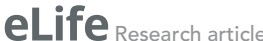

**Figure 3.** BDQ activates the lysosomal pathway in human MTB-infected macrophages. (A) Heatmap showing differential expression of genes included in the Lysosome KEGG category (p-value<0.05). Each column corresponds to one donor. Data were normalized to determine the log ratio with respect to the median expression of each gene. (B) Macrophages were infected with BDQr-MTB expressing the GFP protein and incubated with BDQ (5 μg/mL) for 3 hr, 18 hr and 48 hr. Acid organelles were then labeled with 100 nM LysoTracker DND-99 for 1 hr. The fluorescence intensity was quantified by flow cytometry. (C–E) Cells were infected with GFP expressing BDQr-MTB (green) and treated with BDQ (5 μg/mL). After 18 hr and 48 hr of treatment, cells were labelled with LysoTracker (red) and fluorescence was analyzed by confocal microscopy. DAPI (blue) was used to visualize nuclei (scale bar: 10 μm). The quantification of LysoTracker staining and the percentage of LysoTracker-positive MTB phagosomes were performed using Icy software. (F) Macrophages were activated with heat-killed MTB and treated with BDQ for 18 hr and 48 hr. Cells were then incubated with DQ-Green BSA. Fluorescence was quantified by flow cytometry. Significant differences between BDQ treatment and control (DMSO) are indicated by an asterisk. One representative experiment (of at least three) is shown. Error bars represent the mean ± SD. *p<0.05, **p<0.01, ***p<0.001.

The online version of this article includes the following figure supplement(s) for figure 3:

**Figure supplement 1.** BDQ activates the lysosomal pathway in human naive macrophages.
**Figure supplement 2.** BDQ activates the lysosomal pathway, regardless of infection with live MTB.

revealed the appearance of numerous acidic compartments upon treatment (*Figure 3C*), up to five times more in BDQ-treated macrophages than untreated cells at 48 hr post-treatment (p<0.001, *Figure 3D*). We also observed a large number of MTB phagosomes co-localized with LysoTracker-positive compartments (*Figure 3E*). As the expression of many genes coding for hydrolases was upregulated upon BDQ treatment (*Figure 3A*), we tested the effect of the drug on late endosomal/lysosomal proteolytic activity. BDQ-treated macrophages were incubated with DQ-Green BSA, a self-quenched non-fluorescent probe that produces brightly fluorescent peptides following hydrolysis by lysosomal proteases. At 18 hr and 48 hr post-treatment, we observed a dose-dependent increase in fluorescence intensity upon treatment with BDQ (up to 5.5 times more than untreated cells, p<0.01, *Figure 3F*). Similar results were obtained when we incubated naive macrophages with BDQ (*Figure 3—figure supplement 1*). Together, these data demonstrate that BDQ induces biogenesis of competent lysosomes.

We performed additional experiments to confirm that the main effects of BDQ on lysosome were independent of infection with live bacteria. Briefly, cells were untreated or stimulated with LPS (TLR4 agonist), Pam3CSK4 (TLR1/2 agonist), heat-killed bacteria (hk-MTB), drug-susceptible MTB or BDQr-MTB, and treated with BDQ. After 18 hr, RNA was collected and we performed RT-qPCR on a panel of lysosomal genes. We also analyzed the intensity of the LysoTracker staining using flow cytometry (*Figure 3—figure supplement 2*). Our results clearly show that the main effects on lysosome biogenesis/activation occurred with BDQ treatment and were not exclusively seen after infection with live MTB.

## BDQ potentiates PZA antimycobacterial activity

The capacity of BDQ to induce acidic compartments may potentiate the efficacy of other drugs, whose activity is pH dependent. In vivo studies have suggested a synergistic interaction between BDQ and PZA (*Ibrahim et al., 2007*), and it is commonly assumed that a low pH is required for PZA activity against MTB (*Zhang and Mitchison, 2003*). We thus infected macrophages with BDQr-MTB and treated them with BDQ and PZA. After 7 days of treatment, cells were lysed and bacteria counted. PZA showed moderate bactericidal activity, with 50 µg/mL PZA resulting in a 36% decrease in bacterial numbers compared to untreated cells (*Figure 4A*). We confirmed that the combination of PZA with BDQ was highly bactericidal on MTB, leading to a 83% decrease in colony forming units using 50 µg/mL PZA. This decrease was not a result of an additive effect between the two drugs, as BDQ alone at 1 µg/mL had no antibacterial activity. We also found no synergy between BDQ and

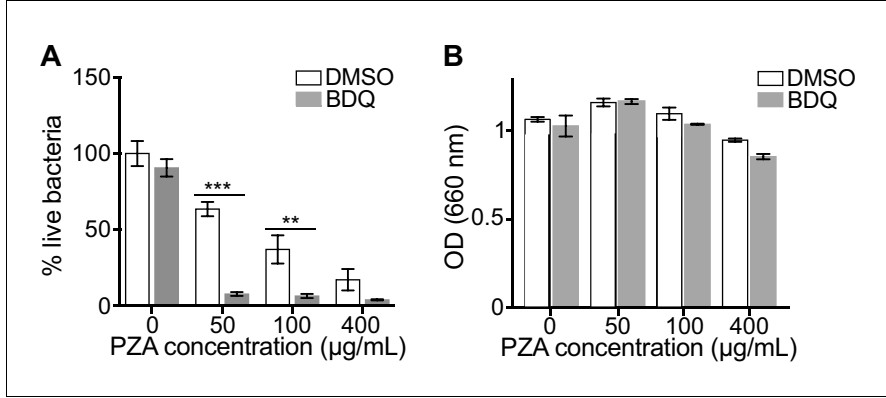

**Figure 4.** BDQ potentiates PZA antimycobacterial activity. (**A**) Macrophages were infected with BDQr-MTB and treated with BDQ (1 µg/mL) and PZA. After 7 of days treatment, cells were lysed and bacteria were enumerated by CFU (counted in triplicate). (**B**) Optical density (OD) measurements of bacterial growth of BDQr-MTB in the presence of BDQ (1 µg/mL) and different concentrations of PZA. Bacteria were cultured in 7H9 medium supplemented with 10% OADC enrichment with/without the drugs. One representative experiment (of at least three) is shown. Error bars represent the mean ± SD. *p<0.05, **p<0.01, ***p<0.001.

The online version of this article includes the following figure supplement(s) for figure 4:

**Figure supplement 1.** BDQ did not potentiate the activity of the other first-line anti-TB drugs, in liquid culture or in BDQr-MTB infected macrophages.

PZA on the BDQ-resistant mutant cultivated in Middlebrook 7H9 liquid medium (*Figure 4B*). Thus, the potentiation of PZA activity by BDQ is most likely due to the effect of BDQ on the host cell, and in particular on the increase of lysosomal acidification.

We next tested whether BDQ would synergize with the other first-line anti-TB drugs in liquid culture or in BDQr-MTB-infected macrophages. We found that BDQ did not potentiate the activity of ethambutol (EMB), isoniazid (INH) and rifampicin (RIF) in either case. While we cannot exclude the possibility that BDQ may have additive effects with other anti-TB drugs, as has been described (*Ibrahim et al., 2007*), we saw no evidence of synergism with any of the first-line agents (*Figure 4— figure supplement 1*).

## Classical anti-TB drugs did not activate the lysosomal pathway in human macrophages

Bacterial hydroxyl radical released after treatment with INH or PZA can directly induce host cell autophagy (*Kim et al., 2012*). We thus tested whether other antibiotics might have similar effects to BDQ. We characterize the genome-wide gene expression profiles of naïve macrophages and macrophages stimulated with hk-MTB, and treated with amikacin (AMK), EMB, INH, PZA or RIF for 18 hr. We chose drug concentrations based on the concentrations detected in the plasma of treated TB patients. Following treatment, only RIF and PZA significantly modulate gene expression in macrophages. 556 and 752 genes were differentially expressed in cells stimulated with heat-killed bacteria and exposed to RIF and PZA, respectively (*Figure 5A*, *Figure 5—figure supplement 1*, *Figure 5— source datas 1–5*). We classified these genes by performing gene-set enrichment analysis and confirmed that the lysosomal pathway was not induced upon RIF or PZA treatment (*Table 2*). The expression of only two genes belonging to this pathway was upregulated by RIF, and only one by PZA, compared to 46 whose expression was modulated by BDQ (*Figure 5B–C*). Consistent with these results, none of these antibiotics were able to increase LysoTracker staining (*Figure 5D*).

## BDQ induces autophagy activation in macrophages

Given BDQ's effect on lysosomal acidification we asked whether it promoted lysosome formation. Lysosome biogenesis is linked to the endocytic and autophagic pathways. Autophagy delivers cytoplasmic material and organelles for lysosomal degradation and is implicated in the immune response to microbes (*Germic et al., 2019*). We therefore tested three inhibitors of the autophagy pathway on BDQ activity: bafilomycin (BAF), which inhibits the V-ATPase; chloroquine (CQ), a lysomotropic agent which prevents endosomal acidification and impairs autophagosome fusion with lysosomes; and 3-methyladenine (3-MA) which blocks autophagosome formation by inhibiting of the type III phosphatidylinositol 3-kinases (PI-3K). We infected macrophages with BDQr-MTB and incubated the cells with BDQ in the presence or absence of the different inhibitors. These molecules were not toxic at the concentrations tested (*Figure 1—figure supplement 1B*). After 2 days, we analyzed Lyso-Tracker staining as a read-out of lysosome activation using flow cytometry and observed that all three inhibitors prevented the increase in staining upon BDQ treatment (*Figure 6A*).

Microtubule-associated protein light-chain 3B (LC3B) is involved in the formation of autophagosomes and autolysosomes. We observed an increase of LC3B puncta per cell at 18 hr and 48 hr post-BDQ treatment using confocal microscopy (*Figure 6B–C*), which was associated with the detection of lipidated LC3 (LC3-II), the form of LC3 recruited to autophagosomal membranes, and with a decrease in sequestosome 1 (SQSTM1) or p62 levels (*Figure 6D–E*). p62 is a ubiquitin-binding scaffold protein, which is degraded upon autophagy induction, and which is used as a marker of autophagic flux (*Liu et al., 2016*). Given we have previously observed that some mycobacterial phagosomes colocalized with lysosomes in BDQ-treated cells (*Figure 3E*), we tested whether BDQ promotes MTB killing, independently of its bactericidal activity on MTB by autophagy. BDQ significantly reduced the number of bacteria (measured by CFU) in cells infected with BDQ-resistant MTB. This effect was completely inhibited by the autophagy inhibitors 3-MA and BAF (*Figure 6F*). Overall, these data show BDQ activates the autophagy pathway in human macrophages and this is involved in its anti-TB activity.

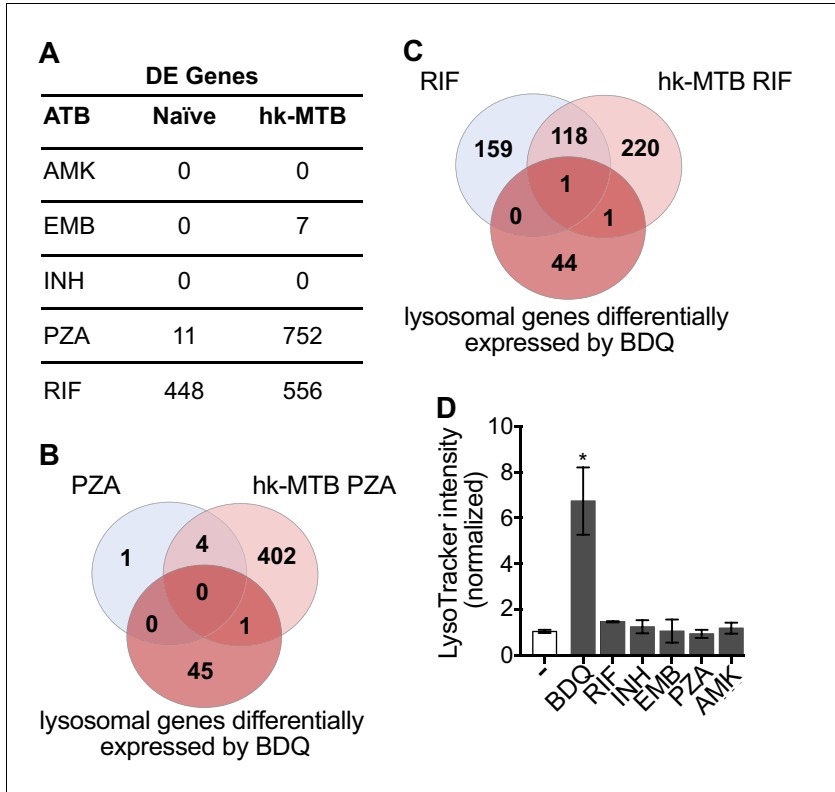

**Figure 5.** Classical anti-TB drugs did not activate the lysosomal pathway in human macrophages. (**A**) Number of differentially-expressed genes upon treatment with commonly used anti-TB drugs relative to untreated control. Briefly, naïve- and hk-MTB-stimulated macrophages were treated with AMK (20 µg/mL), EMB (20 µg/mL), INH (10 µg/mL), PZA (200 µg/mL) or RIF (20 µg/mL). After 18 hr, differentially expressed (DE) genes were identified by mRNAseq. (**B–C**) Venn diagram showing the number of genes regulated by PZA and RIF in naive and hk-MTB-stimulated macrophages, in comparison with the number of lysosomal genes differentially expressed by BDQ (FDR < 0.05, absLogFC >0.1). (**D**) Macrophages were incubated for 48 hr with AMK, BDQ, EMB, INH, PZA or RIF, and then stained with Lysotracker. Fluorescence intensity was analyzed by flow cytometry. One representative experiment (of at least three) is shown. Error bars represent the mean ± SD. *p<0.05.

The online version of this article includes the following source data and figure supplement(s) for figure 5:

**Source data 1.** Differentially expressed genes in hk-MTB stimulated macrophages upon EMB treatment.
**Source data 2.** Differentially expressed genes in hk-MTB stimulated macrophages upon RIF treatment.
**Source data 3.** Differentially expressed genes in naive macrophages upon RIF treatment.
**Source data 4.** Differentially expressed genes in hk-MTB stimulated macrophages upon PZA treatment.
**Source data 5.** Regulated genes naive macrophages upon PZA treatment.
**Figure supplement 1.** Scatterplot depicting the correlation between the log2 fold changes (log2FC) in gene expression levels using live MTB and heat-killed bacteria.

## BDQ activates macrophage bactericidal functions

Autophagy plays numerous roles in innate immunity and in host defenses against intracellular pathogens, including MTB (*Gutierrez et al., 2004*). We thus asked if BDQ conferred protection to bacterial infections naturally resistant to BDQ. To test this hypothesis, we infected macrophages with two different bacterial species: a gram-positive bacterium, *Staphylococcus aureus* and a gram-negative bacterium, *Salmonella* Typhimurium. We confirmed that these two species are resistant to BDQ, even when exposed to high concentration of the drug (20 µg/mL, *Figure 7A*). However, when macrophages were incubated with BDQ and then infected with *S. aureus* and *S.* Typhimurium for 24 hr, we observed a substantial decrease in bacterial survival rates (*Figure 7B*).

To determine if autophagy is involved in this antibacterial activity, we incubated the infected cells with the autophagy inhibitor, 3-MA, and were unable to revert the macrophage resistance to *S. aureus* infection upon BDQ treatment (*Figure 7C*). Macrophages are professional phagocytes, which

**Table 2.** Gene Ontology (GO) functional annotation of differentially expressed genes in naïve- or hk-MTB-stimulated cells treated with PZA or RIF.

**hk-MTB + PZA**

| GO category | avg. LogFC | p-value |
| --- | --- | --- |
| Integral to lumenal side of endoplasmic reticulum membrane | 0.16 | 2.79E-04 |
| Cytokine-mediated signaling pathway | 0.27 | 3.28E-04 |
| Interferon-gamma-mediated signaling pathway | 0.21 | 3.56E-04 |
| MHC class I receptor activity | 0.16 | 7.82E-04 |
| Cytosolic small ribosomal subunit | 0.26 | 7.98E-03 |
| MHC class I protein complex | 0.16 | 1.04E-03 |
| Regulation of immune response | 0.22 | 1.33E-03 |
| Negative regulation of MAPK cascade | 0.20 | 1.34E-03 |

**Naïve + RIF**

| GO category | avg. LogFC | p-value |
| --- | --- | --- |
| Mitotic cell cycle | -0.37 | 1.53E-17 |
| DNA replication | -0.34 | 2.59E-13 |
| Cell cycle checkpoint | -0.36 | 4.98E-10 |
| S phase of mitotic cell cycle | -0.32 | 6.29E-10 |
| DNA strand elongation involved in DNA replication | -0.33 | 1.88E-09 |
| G1/S transition of mitotic cell cycle | -0.36 | 2.02E-09 |
| Cell division | -0.34 | 3.96E-09 |
| Cell cycle | -0.32 | 2.26E-07 |

**hk-MTB + RIF**

| GO category | avg. LogFC | p-value |
| --- | --- | --- |
| Melanosome | 0.26 | 4.09E-07 |
| Endoplasmic reticulum unfolded protein response | 0.29 | 7.95E-07 |
| Electron carrier activity | 0.23 | 9.11E-07 |
| Tissue regeneration | -0.23 | 1.63E-05 |
| Response to drug | -0.29 | 2.32E-05 |
| NADP binding | 0.22 | 3.07E-05 |
| Cellular lipid metabolic process | -0.19 | 4.05E-05 |
| Lipid metabolic process | -0.23 | 4.98E-05 |

have evolved a myriad of defense strategies to contain and eradicate bacteria, such as radical formation, phagosome maturation, and metal accumulation (*Weiss and Schaible, 2015*). Upon incubation with BDQ, we detected an increase in the amount of $NO_2-$, a stable derivative of NO, in the culture supernatant of macrophages (*Figure 7D*). When the cells were treated with N(G)-nitro-L-arginine methyl ester (L-NAME), an inhibitor of nitric oxide (NO) synthesis, *S. aureus*-infected cells were unable to effectively control infection upon incubation with BDQ (*Figure 7E*). Thus, our results suggest that BDQ confers innate resistance to bacterial infection through different mechanisms.

## Mitochondrial functions are not affected by BDQ

BDQ affects cardiac electrophysiology by prolonging the QT interval (*TMC207-C208 Study Group et al., 2014*) and it has been suggested that BDQ inhibits the cardiac potassium channel protein encoded by the human ether-a-go-go-related gene (*hERG*) (*Janssen Therapeutics, 2012*). Therefore, to further understand the molecular mechanisms underpinning macrophage activation by BDQ,

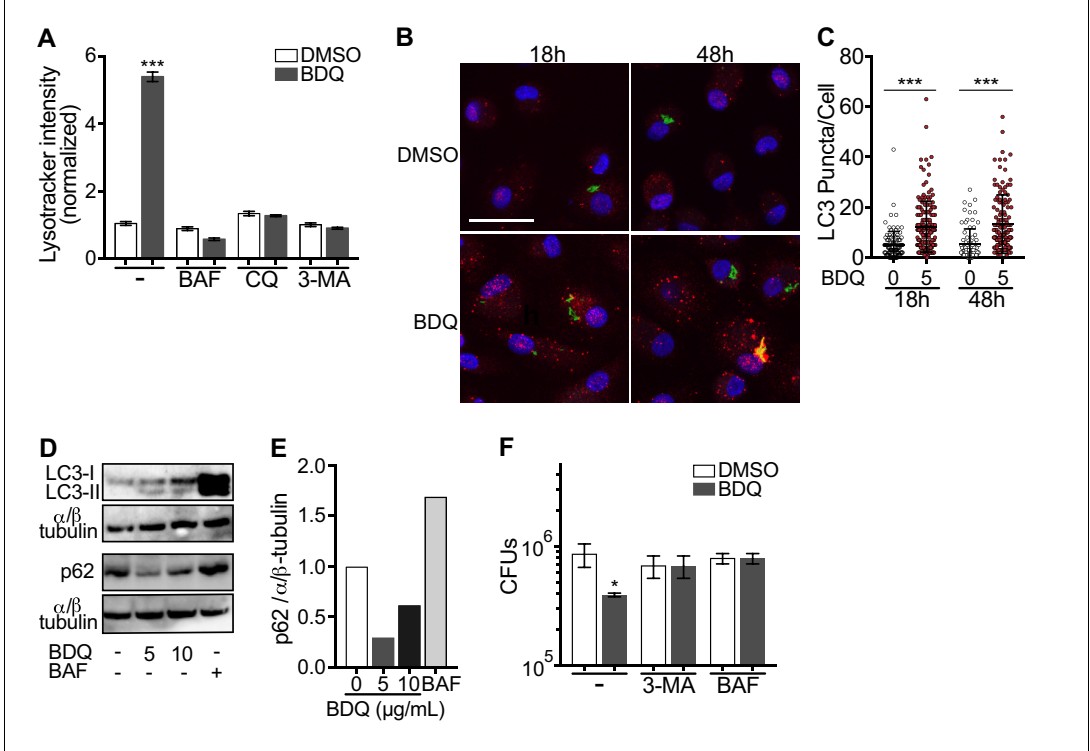

**Figure 6.** BDQ induced autophagy in MTB-infected macrophages. (**A**) BDQr-MTB-infected macrophages were incubated with BDQ (5 µg/mL) and different inhibitors of autophagy; bafilomycin (BAF, 100 nM), chloroquine (CQ, 40 µM) and 3-methyladenine (3-MA, 5 mM). After 48 hr, acidic compartments were stained with LysoTracker and fluorescence quantified by flow cytometry. (**B**) Detection by indirect immunofluorescence of LC3 (red) in BDQr-MTB (green) infected macrophages, treated with BDQ for 18 hr and 48 hr (scale bar: 10 µm). DAPI (blue) was used to visualize nuclei. (**C**) Determination of the number of LC3-positive puncta per cell (one-way ANOVA test). (**D**) Western blot analysis of LC3, p62, and α/ß-tubulin in MTB-infected cells treated with BDQ and BAF. (**E**) Densitometric quantification of p62 staining. (**F**) BDQr-MTB-infected macrophages were left untreated or incubated with BDQ, 3-methyladenine (3-MA) and/or bafilomycin (BAF). After 48 hr, the number of intracellular bacteria was enumerated. One representative experiment (of three) is shown. Error bars represent the mean ± SD. *p<0.05, **p<0.01, ***p<0.001.

we determined if human monocyte-derived macrophages expressed hERG, but were unable to detect *hERG* RNA by RT-qPCR (*Figure 8—figure supplement 1*).

We investigated if BDQ might interfere with other activities of mitochondria. Conflicting reports suggest that BDQ inhibits the mitochondrial ATPase (*Fiorillo et al., 2016*; *Haagsma et al., 2009*). We have already shown that there were no significant differences in the amount of cardiolipin, a constituent of inner mitochondrial membranes, between BDQ-treated cells and control cells (*Figure 2C*). We quantified changes in mitochondrial membrane potential using flow cytometry in cells incubated with BDQ or with oligomycin, a positive control, which hyperpolarizes the mitochondrial membrane potential, and stained with TMRM. TMRM is a fluorescent cell-permeant dye that accumulates in active mitochondria with intact membrane potentials. No changes were observed when macrophages were incubated with the BDQ for 6, 24 and 48 hr (*Figure 8A*). We obtained similar results when mitochondria were stained with MitoTracker Red FM whose accumulation in mitochondria is dependent upon membrane potential (*Figure 8B*). We also measured the oxygen consumption rate (OCR), and detected no change in basal respiration, ATP-linked respiration, maximal respiration, and non-mitochondrial respiration in cells treated with BDQ for 24 hr and 48 hr as compared to untreated cells (*Figure 8C*, *Figure 8—figure supplement 2*).

Mitochondrial reactive oxygen species (ROS) are involved in the regulation of several physiological and pathological processes, including autophagy (*Sena and Chandel, 2012*). We thus stained for mitochondrial superoxide using the MitoSOX dye in BDQ-stimulated cells. Again, we saw no difference upon antibiotic treatment (*Figure 8D*). Incubation with the antioxidant glutathione (GSH) or with its precursor N-Acetyl cysteine (NAC), which prevent the formation of mitochondrial ROS and



**Figure 7.** BDQ increases macrophage bactericidal functions. (A) Growth of *S.* Typhimurium and *S. aureus* in liquid medium in the presence of BDQ (20 µg/mL). (B) Macrophages were incubated with BDQ and then infected with *S.* Typhimurium or *S. aureus*. The number of intracellular bacteria was enumerated at 24 hr post-infection. (C) BDQ-treated macrophages were incubated with 3-MA and then infected with *S. aureus*. The number of bacteria was counted as previously. (D) Quantification of NO2- in the supernatant of macrophages incubated with BDQ for 18 hr and 48 hr. (E) Cells were treated as in (C), 3-MA was replaced by L-NAME (0.1 mM), an inhibitor of nitric oxide (NO) synthesis. One representative experiment (of three) is shown. Error bars represent the mean ± SD. Unpaired two-tailed Student's t test was used. *p<0.05, **p<0.01, ***p<0.001.

reactive nitrogen species (RNS), did not prevent lysosome activation and the killing of *S. aureus* by BDQ (*Figure 8E*). Based on these results, it is unlikely that BDQ alters mitochondrial function in human macrophages.

## BDQ regulates lysosome activation through TFEB and calcium signaling

Given that BDQ induced a lysosomal gene expression signature in macrophages, we wondered whether BDQ could activate the basic helix-loop-helix transcription factor EB (TFEB). TFEB is a master regulator of autophagy and lysosome biogenesis (*Settembre et al., 2011*). In resting cells, TFEB is largely cytosolic and inactive, but upon activation, it translocates into the nucleus and activates the transcription of many autophagy and lysosomal genes (*Settembre et al., 2011*). We therefore analyzed the cellular localization of TFEB, using confocal microscopy. At 18 hr post-treatment, TFEB was mainly localized in the nucleus of BDQ-treated cells (*Figure 9A–B*). The activity of TFEB is regulated by phosphorylation on specific amino acid residues, and its activation is mediated by calcineurin, an endogenous serine/threonine phosphatase, through $Ca^{2+}$ release from the lysosome (*Medina et al., 2015*). In agreement with these studies, we observed an increase in intracellular $Ca^{2+}$ concentration in macrophages treated for 18 hr with BDQ (*Figure 9C*), and confirmed that this intracellular calcium accumulation was required for antibiotic-induced TFEB translocation to the nucleus and lysosomal

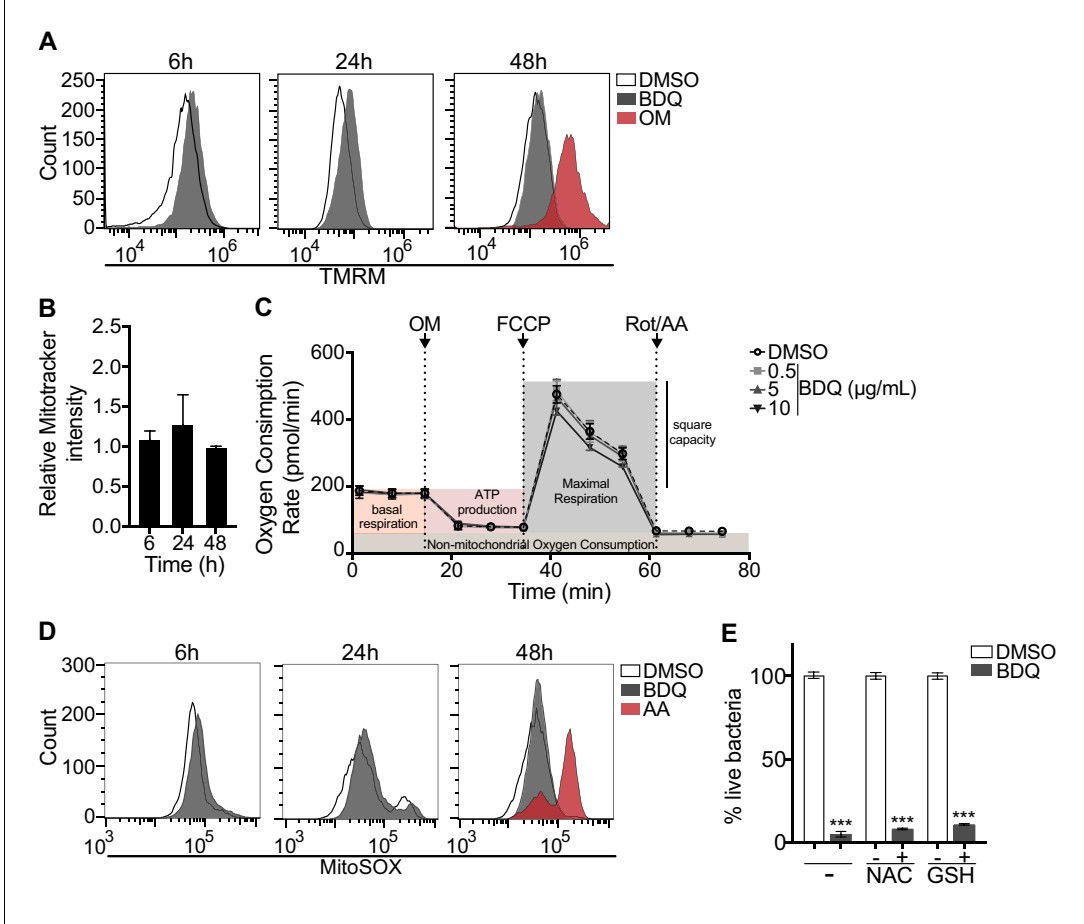

**Figure 8.** BDQ does not affect mitochondrial functions. (**A**) Macrophages were treated with BDQ for 6, 24 and 48 hr and incubated with TMRM. Fluorescence was quantified by flow cytometry. Oligomycin (OM) is a positive control. (**B**) Cells were treated as in (**A**), and mitochondria were stained with MitoTracker Red FM. The graph represents the ratio between the mean fluorescence intensity of BDQ-treated cells and DMSO-treated cells. (**C**) Oxygen consumption rate (OCR) measured by Seahorse extracellular flux assay of cells incubated with BDQ for 24 hr. Basal respiration, ATP production, maximal respiration, respiratory reserve, and nonmitochondrial respiration were followed by sequential additions of oligomycin (OM, an inhibitor of the ATPase), the mitochondrial oxidative phosphorylation uncoupler FCCP, and the inhibitors of electron transport rotenone/antimycin A (Rot/AA). Error bars represent the mean ± SD of three technical replicates. One representative experiment (of two) is shown. (**D**) At each indicated time points, mitochondrial superoxide (mROS) were stained using the MitoSOX dye in BDQ-treated cells. Antimycin A (AA) is used as a positive control. (**E**) Cells were incubated with BDQ in the presence or not of two antioxidants, glutathione (GSH) and its precursor N-Acetyl-L-cysteine (NAC). After 24 hr, the cells were infected with *S. aureus* for an additional 24 hr. Macrophages were lysed and the number of intracellular bacteria enumerated. One representative experiment (of three) is shown. Error bars represent the mean ± SD. was used. ***p<0.001.
The online version of this article includes the following figure supplement(s) for figure 8:

**Figure supplement 1.** The *hERG* gene is not expressed in human monocyte-derived macrophages.
**Figure supplement 2.** Oxygen consumption rate (OCR) measured by Seahorse extracellular flux assay of cells incubated with BDQ for 48 hr.

gene expression. Upon treatment with BAPTA-AM, a cell permeable $Ca^{2+}$ chelator, TFEB remained localized in the cytoplasm of BDQ-treated cells (*Figure 9D*), and we were unable to detect changes in the expression of a panel of lysosomal genes, previously identified as differentially expressed in macrophages incubated with BDQ (*Figure 9E*). The increased bactericidal activity against *S. aureus* was also abrogated in the presence of BAPTA-AM (*Figure 9F*).

To study the role of TFEB in the enhancement of bactericidal activity upon BDQ treatment in more depth, we inactivated TFEB expression in human macrophages using siRNA-mediated gene silencing (*Troegeler et al., 2014*). The level of TFEB expression was decreased by about 85% upon silencing (*Figure 9—figure supplement 1A*). After treating the cells with BDQ, we analyzed the intensity of the LysoTracker staining using flow cytometry and performed RT-qPCR on a panel of



**Figure 9.** Activation of TFEB by BDQ. (**A**) Representative fluorescence microscopy images of macrophages treated with BDQ for 6 hr and 18 hr, or incubated in HBSS for 1 hr (starvation). Cells were stained with antibody against TFEB (red). DAPI (white) was used to visualize nuclei. Scale bar: 10 μm. (**B**) Ratio between nuclear and cytosolic TFEB fluorescence intensity (n > 100 cells per condition, two-way ANOVA test). (**C**) Macrophages were treated

*Figure 9 continued on next page*

*Figure 9 continued*

with BDQ for 18 hr and loaded with the fluorescent calcium binding dye Fluo-8 AM. After 1 hr of incubation, Ca2+ concentration was monitored by FLUOstar Omega. (D) Ratio between nuclear and cytosolic TFEB fluorescence intensity in starved cells and in cells treated with BDQ and/or with the intracellular calcium chelator BAPTA-AM (n > 100 cells per condition, two-way ANOVA test). (E) Relative gene expression measured by RT-qPCR for a panel of differentially expressed lysosomal genes. BDQ-treated macrophages were either left untreated or incubated with BAPTA-AM. Relative expression levels were normalized to the *rpl24* gene. (F) Macrophages were treated with BDQ with or without BAPTA-AM, and then infected with *S. aureus*. After 1 day, cells were lysed and the number of intracellular bacterial colonies was counted (unpaired two tailed Student's t test). (G) TFEB expression was inhibited in macrophages using siRNA-mediated gene silencing. Cells were then treated with BDQ. After 48 hr of treatment, Lysotracker staining was quantified by flow cytometry. (H) Relative gene expression measured by RT-qPCR for a panel of lysosomal genes in TFEB-silenced cells. Relative expression levels were normalized to the *rpl24* gene. (I) TFEB-silenced macrophages were treated with BDQ and infected with *S. aureus*. After 1 day, the cells were lysed and the number of intracellular bacterial colonies was counted. (J) TFEB-silenced macrophages were treated with BDQ and infected with BDQr-MTB. After 48 hr, the number of bacteria was counted. Error bars represent the mean ± SD. was used. *p<0.05, **p<0.01, ***p<0.001.

The online version of this article includes the following figure supplement(s) for figure 9:

**Figure supplement 1.** BDQ-induced lysosome activation is dependent on TFEB.

lysosomal genes. In TFEB-silenced cells, BDQ increases slightly the LysoTracker staining (*Figure 9G*), and the expression of *atp6v0a1, mcoln1, scarb1 or trpm2* (*Figure 9H*). We also infected macrophages with *S. aureus* and MTB, and showed that that TFEB activation is required for better control of both pathogens upon BDQ treatment (*Figure 9I–J*). We next evaluated the recruitment of TFEB to the promoters of some lysosomal genes. As expected, BDQ induces the recruitment of TFEB to these promoters (*Figure 9—figure supplement 1B*). Collectively, our data indicate that BDQ activates TFEB in macrophages and in this way modulates innate immune resistance to bacterial infection.

## Discussion

The emergence of bacterial strains resistant to antibiotics requires the constant development of new antibiotics, which, beyond their bactericidal activity, may have a significant impact on cellular functions. Here, we have analyzed the effects of the new anti-TB drug BDQ on human macrophages. We found that in addition to its antibacterial activity, BDQ induces cell reprogramming, increasing macrophage bactericidal activity. Gene expression profiling revealed that 1495 genes were differentially expressed in MTB-infected macrophages incubated with BDQ, with over-representation of genes involved in metabolism, lysosome biogenesis, and acidification. Recent work has highlighted the role of metabolic reprogramming in controlling immunological effector functions, emphasizing the close connection between cell function and metabolism (*Wang et al., 2019*). In agreement with these results, we observed a substantial increase in both the number of acidic compartments and proteolytic activity of macrophages upon BDQ treatment.

BDQ is a cationic amphiphilic drug, consisting of a hydrophobic ring structure and a hydrophilic side chain with a charged cationic amine group (*Diacon et al., 2012*). Cationic amphiphilic drugs can accumulate in lysosomes through ion trapping (*de Duve et al., 1974*). At neutral pH, they passively diffuse across cell and organelle membranes but when they enter the luminal space of acidic compartments such as lysosomes, the amine group ionizes and becomes membrane-impermeable (*MacIntyre and Cutler, 1988*). Such lysosomotropic compounds usually increase the lysosomal pH and thus decrease lysosomal enzyme activity (*Kazmi et al., 2013*). However, our results reveal instead that BDQ triggers lysosomal activation, upregulating the expression of genes coding for hydrolases and for subunits of the lysosomal proton pump v-ATPase. Consistent with these observations, we observed that BDQ-treated cells significantly increase their ability to degrade DQ-BSA.

Pre-clinical studies have shown that BDQ may induce phospholipidosis, potentially explaining some of the drug's observed toxicities (*Diacon et al., 2012*). Phospholipidosis, which is characterized by the accumulation of phospholipids in lysosomes, resulting in impaired lysosome function, is common upon treatment with cationic amphiphilic compounds (*Shayman and Abe, 2013*). Various phospholipid species have been described including phosphatidylcholine, phosphatidylethanolamine, phosphatidylserine, and lysobisphosphatidic acid (*Reasor, 1984*; *Yamamoto et al., 1971a*; *Yamamoto et al., 1971b*; *Yoshikawa, 1991*). In BDQ-treated macrophages, we only observed an

increase in the amount of phosphatidylinositol and phosphatidylinositol-4-phosphate. The quantity of cardiolipin, phosphatidylethanolamine, and phosphatidylglycerol remained unchanged upon treatment. These observations do not indicate lysosomal dysfunction, but rather a targeted regulation of certain phospholipids by BDQ. In accordance with this idea, 28 genes involved in phospholipid metabolism were differentially expressed in BDQ-treated macrophages. Phosphatidylinositol phosphates regulate many cellular functions, including endosomal trafficking, endoplasmic reticulum (ER) export, autophagy, and phagosome-lysosome fusion (*De Matteis et al., 2013*; *Levin et al., 2017*). These phospholipids may thus be involved in the increase of autophagy and mycobacterial phagosome-lysosome fusion upon BDQ treatment.

Lysosomes are both digestive organelles of the endocytic and autophagic pathways and signaling hubs involved in nutrient sensing, cell growth and differentiation, transcriptional regulation, and metabolic homeostasis (*Lamming and Bar-Peled, 2019*; *Lawrence and Zoncu, 2019*). In response to nutrients and growth factors, the mechanistic target of the rapamycin complex 1 (mTORC1) is recruited and activated at the lysosomal surface, where it promotes ribosomal biogenesis, translation, and biosynthesis of lipids (*Lamming and Bar-Peled, 2019*; *Lawrence and Zoncu, 2019*). mTORC1 binds to and phosphorylates TFEB, resulting in its cytosolic sequestration (*Roczniak-Ferguson et al., 2012*; *Settembre et al., 2012*). Upon starvation or lysosomal stress, mTORC1 is released from the lysosomal membrane and becomes inactive (*Lamming and Bar-Peled, 2019*; *Lawrence and Zoncu, 2019*). The release of lysosomal $Ca^{2+}$ activates the phosphatase calcineurin, which de-phosphorylates TFEB and promotes its nuclear translocation (*Medina et al., 2015*). TFEB then binds to CLEAR (coordinated lysosomal expression and regulation) elements within the promoters of genes involved in autophagy and lysosomal biogenesis and activates their expression (*Lamming and Bar-Peled, 2019*; *Lawrence and Zoncu, 2019*). We found that TFEB translocates from the cytoplasm to the nucleus in BDQ-treated cells, with the concomitant up-regulation of 85 genes containing CLEAR elements 18 hr after incubation with the drug.

Finding the direct host-cell target of BDQ would have been a major breakthrough, but unfortunately, we were unable to identify the target. However, it is possible that BDQ could modulate the host response by interfering with ion homeostasis inside the lysosome. Greenwood et al. recently showed that BDQ accumulated primarily in host cell lipid droplets (*Greenwood et al., 2019*). It is tempting to speculate that these droplets are actually lysosomes. Due to its high hydrophobicity, BDQ might accumulate in lysosomal membranes and thereby change the transmembrane ion permeability. In agreement with this hypothesis, it has been shown that BDQ can accumulate at the lipid membrane of liposomes and act as a $H^+/K^+$ ionophore (*Hards et al., 2018*). The ensuing lysosomal stress could then facilitate the dissociation of mTOR from the lysosomal membrane (*Plescher et al., 2015*) and the activation of TFEB. Interestingly, BDQ increases the expression of mucolipin 1 gene (*mcoln1*, **Figure 3A**, **Figure 3—figure supplement 2A**), which in turn can activate TFEB. Medina et al. showed indeed that lysosomal $Ca^{2+}$ is released through MCOLN1 and activates calcineurin, which binds and dephosphorylates TFEB (*Medina et al., 2015*).

A striking feature of BDQ-treated macrophages is their capacity to control pathogenic bacterial infection. BDQ enhances macrophage innate defense mechanisms, including induction of antimicrobial effectors such as nitric oxide, phagosome-lysosome fusion, and autophagy. Other anti-TB drugs have been described to regulate autophagy. INH and PZA promote autophagy activation and phagosomal maturation in MTB-infected murine macrophages (*Kim et al., 2012*). This process was dependent of bacterial factors and was suggested to be essential for antimycobacterial drug action and for dampening proinflammatory cytokines (*Kim et al., 2012*). In our system, we did not detect increased autophagy in cells treated with INH, which may also be due to differences in the autophagy response in murine and human macrophages. Altogether, we demonstrate that BDQ is able to boost the innate defenses of human cells.

A growing number of pathogenic bacteria are becoming resistant to antibiotics, making their use less effective. In addition to the development of 'classical' drugs targeting key factors in bacterial physiology, host-directed therapy (HDT) has emerged as approach that could be used in adjunct with existing or future antibiotics (*Machelart et al., 2017*). For example, metformin, an FDA-approved drug for type II diabetes, increases the production of mitochondrial reactive oxygen species and stimulates phagosome-lysosome fusion by activating the 5′-adenosine monophosphate-activated protein kinase (AMPK) (*Singhal et al., 2014*), and recent studies suggest that metformin provides better outcomes in TB patients, especially those with diabetes mellitus (*Yew et al., 2019*).

Pathogens manipulate host-signaling pathways to subvert innate and adaptive immunity. It might thus be possible to reprogram the host immune system to better control or even kill bacteria. For instance, MTB has developed several strategies to counteract autophagy, including the product of the enhanced intracellular survival (Eis) gene, which limits ROS generation (*Shin et al., 2010*). Our results clearly show that BDQ can bypass these escape mechanisms and allow more effective control of bacterial infection. We also showed that BDQ potentiates the activity of other anti-TB drugs, independently of its bactericidal activity on MTB. Hence, our work opens new avenues for downstream evaluation of the potential use of BDQ as a potent drug in HDT.

# Materials and methods

## Key resources table

| Reagent type (species) or resource | Designation | Source or reference | Identifiers | Additional information |
|---|---|---|---|---|
| Buffy coats (*Homo sapiens*) | PBMC | Etablissement Français du sang (EFS) | | |
| Cell line (*Homo sapiens*) | HEK 293 cell line | Obtained from Dr. Craig T. January, University of Wisconsin–Madison | | |
| Cell line (*Homo sapiens*) | HEK 293 cell line stably expressing hERG | Obtained from Dr. Craig T. January, University of Wisconsin–Madison | | |
| Strain, strain background (*Mycobacterium tuberculosis*) | H37Rv | PMID:12574362 | | |
| Strain, strain background (*Salmonella* Typhimurium) | *S.* Typhimurium | PMID:25793259 | | |
| Strain, strain background (*Staphylococcus aureus*) | *S. aureus* | PMID:25793259 | | |
| Antibody | Donkey anti-Rabbit Secondary Antibody, Alexa Fluor 555 | Thermo Fisher | Cat. #: A-31572 RRID:AB_162543 | IF (1:500) |
| Antibody | Goat anti-Mouse Secondary Antibody, Alexa Fluor 555 | Thermo Fisher | Cat. #: A-21424 RRID:AB_141780 | IF (1:500) |
| Antibody | Mouse monoclonal anti-LC3 | MBL | Cat. #: M152-3 RRID:AB_1953013 | IF (1:100) |
| Antibody | Rabbit IgG HRP Linked Whole Ab | GE Healthcare | Cat. #: NA934 RRID:AB_2722659 | WB (1:1000) |
| Antibody | Rabbit polyclonal anti- α/β-Tubulin | Cell Signaling Technology | Cat. #: 2148S RRID:AB_2288042 | WB (1:1000) |
| Antibody | Rabbit polyclonal anti-LC3B | Abcam | Cat. #: Ab48394 RRID:AB_881433 | WB (1:1000) |
| Antibody | Rabbit polyclonal anti-SQSTM1/p62 | Cell Signaling Technology | Cat. #: 5114S RRID:AB_10624872 | WB (1:1000) |
| Antibody | Rabbit polyclonal anti-TFEB | Thermo Fisher | Cat. #: PA5-65566 RRID:AB_2662642 | IF (1:100) |

*Continued on next page*

Continued

| Reagent type (species) or resource | Designation | Source or reference | Identifiers | Additional information |
|---|---|---|---|---|
| Commercial assay or kit | Griess Reaction Assay | Promega | Cat. #: G2930 RRID:SCR_006724 | |
| Commercial assay or kit | Illumina TruSeq RNA Sample Preparation kit v2 | Illumina | Cat. #: RS-122-2001/2002 RRID:SCR_010233 | |
| Commercial assay or kit | MTT Cell Proliferation Assay kit | Trevigen | Cat. #: 4890–25K RRID:SCR_012449 | |
| Commercial assay or kit | PierceTM BCA Protein Assay kit | Thermo Fisher | Cat. #: 23227 | |
| Commercial assay or kit | RNeasy Mini kit | Qiagen | Cat. #: 74104 RRID:SCR_008539 | |
| Commercial assay or kit | Seahorse XF Cell Mito Stress Test Kit | Agilent Technologies | Cat. #: 103708–100 | |
| Commercial assay or kit | Seahorse XF Glycolytic rate Assay kit | Agilent Technologies | Cat. #: 103710–100 | |
| Chemical compound, drug | Bedaquiline | Combi-Blocks | Cat. #: QV-7478 | |
| Chemical compound, drug | BAPTA-AM | Sigma-Aldrich | Cat. #: A1076 | |
| Commercial assay or kit | Image-IT TMRM | Thermo Fisher | Cat. #: I34361 | |
| Commercial assay or kit | LysoTracker DND-99 | Thermo Fisher | Cat. #: L7528 | |
| Commercial assay or kit | MitoSOX Red | Thermo Fisher | Cat. #: M36008 | |
| Commercial assay or kit | MitoTrackerTM Deep Red FM | Thermo Fisher | Cat. #: M22426 | |
| Commercial assay or kit | Fluo-8 AM | Abcam | Cat. #: Ab142773 | |
| Commercial assay or kit | DQ Green BSA | Thermo Fisher | Cat. #: D12050 | |
| Commercial assay or kit | SuperScript III Reverse Transcriptase | Thermo Fisher | Cat. #: 18080093 | |
| Commercial assay or kit | Power SYBR Green PCR Master Mix | Thermo Fisher | Cat. #: 4367659 | |
| Sequenced-based reagent | RT-qPCR primers | This paper | | See *Supplementary file 1* |
| Sequenced-based reagent | siRNA: nontargeting control | Dharmacon | Cat. #: D-001810–10-05 | |
| Sequenced-based reagent | siRNA: ON-TARGETplus Human TFEB (7942) siRNA - SMARTpool | Dharmacon | Cat. #: L-009798–00-0005 | |
| Software, algorithm | GraphPad Prism | GraphPad Prism (https://graphpad.com) | RRID:SCR_002798 | Version 7 |
| Software, algorithm | ClueGO (Cytospace plug-in) | http://apps.cytoscape.org/apps/cluego | RRID:SCR_005748 | |

*Continued on next page*

*Continued*

| Reagent type (species) or resource | Designation | Source or reference | Identifiers | Additional information |
|---|---|---|---|---|
| Software, algorithm | DESeq2 | https://bioconductor.org/packages/release/bioc/html/DESeq2.html | RRID:SCR_015687 | Version 1.18.1 |
| Software, algorithm | Icy | http://icy.bioimageanalysis.org/download/ | RRID:SCR_010587 | Version 1.0 |
| Software, algorithm | Seahorse Wave | http://www.agilent.com/en-us/products/cell-analysis-(seahorse)/software-download-for-wave-desktop | RRID:SCR_014526 | |
| Software, algorithm | ImageJ | http://imagej.nih.gov/ij | RRID:SCR_003070 | |

## Ethics statement

Buffy coats were obtained from healthy donors after informed consent. The blood collection protocols were approved by both the French Ministry of Research and a French Ethics Committee. The blood collection was carried out in accordance with these approved protocols by the Etablissement Français du Sang (EFS, n°12/EFS/134).

## Macrophage, MTB and infection

Blood mononuclear cells were isolated from buffy coats by Lymphocytes Separation Medium centrifugation (Eurobio, Les Ulis, France). CD14$^+$ monocytes were isolated by positive selection using CD14 microbeads (Miltenyi Biotec, Bergisch Gladbach, Germany) and were allowed to differentiate into macrophages in the presence of granulocyte macrophage colony-stimulating factor (GM-CSF, 20 ng/mL; Miltenyi Biotec) over a 6-day period. To exclude potential differences due to the MTB bacillary load between treated and untreated cells, macrophages were infected with BDQ-resistant MTB strain H37Rv (BDQr-MTB) expressing green-fluorescent protein (GFP). Briefly, exponentially growing MTB carrying the pEGFP plasmid (*Tailleux et al., 2003*) was plated during 4 weeks on Middlebrook 7H11 agar supplemented with OADC (Becton Dickinson, Franklin Lakes, New Jersey) and containing 0.3 µg/mL BDQ. Some clones were then selected. Resistance to BDQ was confirmed (i) by bacterial culture in Middlebrook 7H9 Broth (Becton Dickinson) supplemented with albumin-dextrose-catalase (ADC, Becton Dickinson) and 0.3 µg/mL BDQ, and (ii) by confirming the mutation in the ATP synthase gene. The *atpE* gene was PCR-amplified using primers (forward: 5-TCGTGTTCA TCCTGATCTCCA-3; reverse: 5-GACAATCGCGCTCACTTCAC-3) and the PCR products were sent to Eurofins for sequencing. All the selected mutants carried a mutation in the *atpE* gene as described previously (*Andries et al., 2005*). Only mutant with similar growth rate (in liquid medium and in macrophages) as the wild-type strain has been used for further experiments. Before infection, bacteria were washed and resuspended in 1 mL PBS. Clumps were disassociated by 50 passages through a needle, and then allowed to sediment for 5 min. The density of bacteria in the supernatant was verified by measuring the OD600 and aliquot volumes defined to allow 0.5 bacterium-per-cell infections. After 2 hr of incubation at 37°C, infected cells were thoroughly washed in RPMI 1640 to remove extracellular bacteria and were incubated in fresh medium.

## Cell viability assay

Cell viability was determined using the LDH Cytotoxicity Assay kit or the MTT assay kit (Abcam), according to manufacturer's instructions.

## Resazurin assay determination of the minimal inhibitory concentration (MIC)

The microdilution test was performed in 96-well plates as previously described (*Palomino et al., 2002*). Briefly, BDQr-MTB and H37Rv were cultured in 7H9 liquid medium containing 2-fold dilutions

of BDQ (from 80 to 0.039 µg/mL) during 6 days. The dye resazurin (Sigma) was then added to each well at a final concentration of 0.003%. After 24 hr, the absorbance was measured at 570 nm.

## High-content screening

Mφs ($10^5$ cells/mL) were plated in 96-well tissue culture plates (CellCarrier-96 Ultra Microplates) and were infected with drug-susceptible H37Rv or BDQr-MTB expressing the GFP protein. Cells were then treated with BDQ (5 µg/mL) for 18 hr or 5 days. Cells were fixed with 4% paraformaldehyde for 1 hr at room temperature and were strained with HCS CellMask Blue stain (HCS, Invitrogen Molecular Probes, 2 µg/mL) and Hoechst 33342 (5 µg/mL) at 4°C in PBS. Confocal images were acquired using the automated fluorescence microscope Opera Phenix High Content Screening System (Perkin Elmer Technology, Waltham, Massachusetts), with a 20x air objective. Images were analyzed using Columbus Conductor Database (Perkin Elmer Technologies).

## RNA isolation, library preparation and sequencing

Total RNA from macrophages was extracted using QIAzol lysis reagent (Qiagen, Hilden, Germany) and purified over RNeasy columns (Qiagen). The quality of all samples was assessed with an Agilent 2100 bioanalyzer (Agilent Technologies, Santa Clara, CA) to verify RNA integrity. Only samples with good RNA yield and no RNA degradation (ratio of 28S to 18S,>1.7; RNA integrity number,>9) were used for further experiments. cDNA libraries were prepared with the Illumina TruSeq RNA Sample Preparation Kit v2 and were sequenced on an llumina HiSeq 2500 at the CHU Sainte-Justine Integrated Centre for Pediatric Clinical Genomics (Montreal, Canada).

STAR v2.5.0b (*Dobin et al., 2013*) was used to map RNA-seq reads to the hg38 reference genome and quantify gene expression (option-quantMode GeneCounts) by counting the fragments overlapping the Ensembl genes (GRCh38 v. 83). Differential expression analysis was performed using a generalized linear model with the R Bioconductor package DESeq2 v1.18.1 (*Love et al., 2014*) on the 12,584 genes with at least one count-per-million (CPM) read in at least four samples. The model formula used in DESeq2 (~Donor + Infection + Infection:Donor + Infection:Treatment + Donor:Treatment) contained: the main effects for Donor and Infection, interactions of Donor with Infection and Treatment to adjust for various responses to infection and treatment between donors, and a nested interaction of Infection with Treatment because we were interested in the infection-status-specific treatment effects. The latter was used to extract differentially expressed genes between treated and untreated samples under the infected and uninfected conditions. p-Values were adjusted for multiple comparisons using the Benjamini-Hochberg method producing an adjusted P-value or false-discovery rate (FDR).

Gene ontology (GO) enrichment analyses were performed using the Cytoscape app ClueGO (version 2.5.3) (*Bindea et al., 2009*). The following parameters were used: only pathways with pV ≤0.01, Minimum GO level = 3, Maximum GO level = 8, Min GO family >1, minimum number of genes associated to GO term = 5, and minimum percentage of genes associated to GO term = 8. Enrichment p-values were calculated using a hypergeometric test (p-value<0.05, Bonferroni corrected).

## Measurement of glycolysis

Measurement of glycolysis was done using the Glycolytic rate assay kit (Seahorse, Agilent Technologies), following the manufacturer's protocol. Briefly, cells were seeded in Xe96 plates treated with BDQ for 24 hr. The cells were then incubated in the assay medium (Seahorse XF Base Medium without phenol, 2 mM glutamine, 10 mM glucose, 1 mM pyruvate and 5.0 mM HEPES) at 37°C, during 1 hr. Extracellular acidification rate (ECAR, milli pH/min) and oxygen consumption rate (OCR, pmol/min) were measured using the Seahorse Bioscience XFe96 Analyzer.

## Lipidomic

Cells were treated with BDQ during 18 hr and them lysed in water during 10 min at 37°C. Samples were heated at 90°C during 40 min in order to inactivate MTB, and were then washed three times to remove salts and contaminants that could preclude the analysis. Prior to mass spectrometry analysis, the 2,5-dihydroxybenzoic acid (Sigma-Aldrich, Saint-Louis, Missouri) matrix was added at a final concentration of 10 mg/mL in a chloroform/methanol mixture at a 90:10 (v/v) ratio; 0.4 µL of a cell solution at a concentration of $2 \times 10^5$ to $2 \times 10^6$ cells/mL, corresponding to ~100–1,000 cells per well of

the MALDI target plate (384 Opti-TOF 123 mm ×84 mm, AB Sciex), and 0.6 µL of the matrix solution were deposited on the MALDI target plate, mixed with a micropipette, and left to dry gently. MALDI-TOF MS analysis was performed on a 4800 Proteomics Analyzer (with TOF-TOF Optics, Applied Biosystems, Foster City, California) using the reflectron mode. Samples were analyzed operating at 20 kV in the negative and positive ion mode. Mass spectrometry data were analyzed using Data Explorer version 4.9 from Applied Biosystems.

## Staining and quantification of acidic compartments

Cells were incubated with LysoTracker DND-99 (100 nM; Thermo Fisher, Waltham, Massachusetts) during 1 hr at 37˚C. Cells were then fixed with 4% paraformaldehyde at room temperature (RT) for 1 hr. Fluorescence was analyzed using a CytoFLEX Flow Cytometer (Beckman Coulter, Brea, California). More than 10,000 events per sample were recorded. The analysis was performed using the FlowJo software.

LysoTracker staining was also analyzed using a Leica TCS SP5 Confocal System. Briefly, cells were washed twice with PBS after incubation with LysoTracker DND-99 (1 µM), fixed with 4% paraformaldehyde for 1 hr at RT, stained with DAPI (1 µg/mL, Thermo Fisher) during 10 min mounted on a glass slide using Fluoromount mounting medium (Thermo Fisher). Quantification of LysoTracker staining was performed using Icy software.

## Quantification of lysosomal proteolytic activity

Macrophages were activated with heat-killed MTB and treated with BDQ during 18 hr or 48 hr. Cells were then incubated with DQ-Green BSA (10 µg/mL; Thermo Fisher) for 1 hr at 37˚C. The hydrolysis of the DQ-Green BSA by lysosomal proteases produces brightly fluorescent peptides. Cells were washed and incubated further in culture medium for 3 hr to ensure that DQ BSA had reached the lysosomal compartment. Cells were detached and were fixed with 4% paraformaldehyde and the fluorescence was analyzed using a CytoFLEX Flow Cytometer (Beckman Coulter).

## Determination of bacterial counts

Macrophages were lysed in distilled water with 0.1% Triton X-100. MTB was enumerated as previously described *Tailleux et al. (2003)* and plated on 7H11. CFUs were scored after three weeks at 37˚C. *S. aureus* and *S.* Typhimurium were plated on Luria-Bertani agar and CFUs were counted after 1 day at 37˚C.

## Indirect immunofluorescence

Macrophages (4 × 10⁵ cells/mL) were grown on 12 mm circular coverslips in 24-well tissue culture plates for 24 hr in cell culture medium, followed by BDQ treatment. Cells were fixed with 4% paraformaldehyde for 1 hr at RT, and were then incubated for 30 min in 1% BSA (Sigma-Aldrich) and 0.075% saponin (Sigma-Aldrich) in PBS, to block nonspecific binding and to permeabilize the cells. Cells were incubated with anti-LC3 (MBL, Woburn, MA) during 2 hr at RT. Alternatively, cells were fixed with cold methanol for 5 min, and were then incubated for 10 min in PBS containing 0.5% saponin. Cells were stained with anti-TFEB (Thermo Fisher) overnight at 4˚C. Cells were washed and incubated with Alexa Fluor 555 secondary antibody (Thermo Fisher) for 2 hr. Nuclei were stained with DAPI (1 µg/mL) during 10 min. After labeling, coverslips were set in Fluoromount G medium containing 1 µg/ml 4′,6-diamidino-2-phenylindole (DAPI) (SouthernBiotech, Birmingham, Alabama) on microscope slides. Fluorescence was analyzed using Leica TCS SP5. Quantification of TFEB staining was performed using Icy software. LC3B puncta were analysed by confocal microscopy and quantified using ImageJ. Infected cells were manually segmented, thresholded and puncta counted using Analyze Particles. Dot plots represent the mean values of at least 83 cells from two donors. Error bars depict the SD.

## Quantitative reverse transcription PCR (RT-qPCR)

Reverse transcription of mRNA to cDNA was done using SuperScript III Reverse Transcriptase (Thermo Fisher) followed by amplification of cDNA using Power SYBR Green PCR Master Mix (Thermo Fisher). All primers used in this study are listed in *Supplementary file 1*. Reactions were

performed using a StepOnePlus Real-Time PCR System Thermal Cycling block (Applied Biosystems). The relative gene expression levels were assessed according to the $2^{-\Delta Ct}$ method (*Pfaffl, 2001*).

## Western blot analysis

Cells were lysed with RIPA buffer (Thermo Fisher) containing protease inhibitor cocktails (Roche) and stored at −80°C. Protein concentration was determined using the BCA protein assay kit (Thermo Fisher) according to the manufacturer instructions. 20 µg of total protein were loaded on a NUPAGE 4–12% Bis-Tris polyacrylamide gel (Thermo Fisher) and transferred to PVDF membranes (iBlot, Thermo Fisher). The membranes were blocked with TBS-0.1% Tween20, 5% non-fat dry milk for 30 min at RT and incubated overnight with primary antibodies against α-β-Tubulin, p-62 (Cell Signaling, Danvers, Massachusetts) and LC3 (Abcam, Cambridge, United Kingdom). Membranes were washed in TBS-Tween and incubated with secondary HRP-conjugated antibody (GE Healthcare, Chicago, IL) at RT for 1 hr. Membranes were washed and exposed to SuperSignal West Femto Maximum Sensitivity Substrate (Thermo Fisher). Detection and quantification of band intensities was performed using Azure Imager C400 (Azure Biosystems, Dublin, CA) and ImageJ software (version 1.51).

## Infection *S. aureus* and *S.* Typhimurium

*S. aureus* and *S.* T*yphimurium* were grown in Luria-Bertani broth. Bacteria were washed three times and resuspended in PBS. The density of bacteria was estimated by measuring the $OD_{600}$. Cells were then infected at a multiplicity of infection of 2:1. After 1 hr of infection, cells were extensively washed and incubated for 1 hr in culture medium supplemented with gentamicin (100 µg/mL). After washing, cells were cultured with different concentrations of BDQ and gentamicin (5 µg/mL).

## Measurement of nitric oxide

NO was measured by Griess reaction assay (Promega, Madison, Wisconsin) according to the manufacturer's instructions. Briefly cell culture supernatants were incubated with sulfanilamide solution during 10 min followed by additional 10 min with N-1-napthylethylenediamine dihydrochloride. The absorbance was measured at 520 nm.

## Mitochondrial membrane potential

Cells were stained with Image-IT TMRM (10 nM, Thermo Fischer) during 30 min at 37°C or with Mito-Tracker Deep Red (100 nM, Thermo Fisher) during 45 min at 37°C. Cells were washed in PBS and detached from culture plates with 0.05% Trypsin-EDTA. Fluorescence was analyzed using a CytoFLEX Flow Cytometer (Beckman Coulter).

## Measurement of oxygen consumption

The oxygen consumption rate was measured using the XF Cell Mito Stress Test Kit (Seahorse, Agilent Technologies) according to the manufacturer's protocol. Briefly, cells were seeded in Xe96 plates and treated with BDQ for 24 hr. The test was performed by adding oligomycin (1 µM), FCCP (1 µM), rotenone and antimycin (0.5 µM) at the indicated time points.

## Mitochondrial ROS assay

Cells were incubated with MitoSOX Red (5 µM, Thermo Fisher) during 10 min at 37°C. Cells were washed in PBS and detached from culture plates with 0.05% Trypsin-EDTA. Fluorescence was analyzed using a CytoFLEX Flow Cytometer (Beckman Coulter).

## Calcium measurement assay

Cells were treated with BDQ for 1 to 18 hr, then labeled with Fluo-8 AM (4 µM, Abcam) during 1 hr. Cells were washed twice with PBS and fluorescence was analyzed using FLUOstar Omega (BMG Labtech, Ortenberg, Germany).

## siRNA silencing of TFEB

siRNA transfection was performed as previously described (*Troegeler et al., 2014*). Briefly, macrophages were transfected using the lipid-based HiPerfect system (Qiagen) and an ON-TARGETplus SMARTpool siRNA targeting TFEB (50 nM) and a non-targeting siRNA (scramble) (Dharmacon). After

6 hr, cells were washed and incubated in complete culture medium for an additional 2 days. The inactivation of TFEB was confirmed RT-qPCR at 48 hr post-transfection.

## Chromatin immunoprecipitation assay

The protocol was adopted from *Blecher-Gonen et al. (2013)*. Briefly, cells treated were fixed in 1% formaldehyde during 10 min. Glycine was then added to a final concentration of 0.15 M. Cells were washed with PBS and were lysed with ice-cold RIPA buffer supplemented with protease inhibitors (Roche). The chromatin fraction was sonicated to obtain fragments from 100 to 500 bp and were coupled with DynaBeads protein G coupled to the anti-TFEB antibody (D2O7D, Cell Signaling) overnight at 4°C. Immune complexes were eluted from the beads with 1% SDS in TE. After treatment with RNaseA and proteinase K, protein-DNA cross-links were reversed by adding NaCl 5M and incubated at 65°C overnight. Chromatin immunoprecipitation analysis was performed by qPCR using Power SYBR Green PCR Master Mix (Thermo Fisher). The primers for *gla* and *mcoln1* were used from *Palmieri et al. (2011)*.

## Quantification and statistical analysis

Data are expressed as means ± standard deviations (SD). Statistical analyses were performed with Prism software (GraphPad Software Inc), using the t test and one-way analysis of variance (ANOVA) as indicated in the figure legends. A p value of < 0.05 was considered to be significant.

## Data availability

The raw fastq files of BDQ-treated cells have been deposited in NCBI's Gene Expression Omnibus (*Edgar et al., 2002*) and are accessible through GEO Series accession number GSE133145. The raw fastq files of cells stimulated with heat-killed MTB or treated with different antibiotics are accessible through GEO Series accession numbers GSE143627 and GSE143731.

# Acknowledgements

We thank Olivier Neyrolles and Howard E Takiff for reading the manuscript and helpful discussion. We gratefully acknowledge the UTechS Cytometry and Biomarkers and the UTechS Photonic BioImaging (Imagopole) Citech of Institut Pasteur (Paris, France) as well as the France–BioImaging infrastructure network supported by the French National Research Agency (ANR-10–INSB–04; Investments for the Future) for support in conducting this study, in particular PH. Commere for help with flow cytometry. We also thank Charles Privé (CHU Sainte-Justine Integrated Centre for Pediatric Clinical Genomics, Montreal, Canada) and Mickael Orgeur (Unit for Integrated Mycobacterial Pathogenomics, Institut Pasteur) for their technical support.

# Additional information

### Funding

| Funder | Grant reference number | Author |
|---|---|---|
| European Commission | 604237 | Brigitte Gicquel |
| Institut Pasteur | | Roland Brosch<br>Brigitte Gicquel<br>Ludovic Tailleux |
| Francis Crick Institute | | Maximiliano G Gutierrez |
| Cancer Research UK | FC001092 | Maximiliano G Gutierrez |
| Medical Research Council | FC001092 | Maximiliano G Gutierrez |
| Wellcome | FC001092 | Maximiliano G Gutierrez |
| Engineering and Physical Sciences Research Council | EP/M027007/1 | Gérald Larrouy-Maumus |
| Fondation pour la Recherche Médicale | FDM201806006250 | Alexandra Maure |

| Agence Nationale de la Recherche | ANR-10-LABX-62-IBEID | Roland Brosch |

The funders had no role in study design, data collection and interpretation, or the decision to submit the work for publication.

## Author contributions

Alexandre Giraud-Gatineau, Conceptualization, Formal analysis, Validation, Investigation, Visualization, Methodology, Writing - original draft, Writing - review and editing; Juan Manuel Coya, Formal analysis, Validation, Investigation, Visualization, Methodology; Alexandra Maure, Elliott M Bernard, Jade Marrec, Formal analysis, Investigation; Anne Biton, Michael Thomson, Formal analysis, Investigation, Methodology; Maximiliano G Gutierrez, Gérald Larrouy-Maumus, Funding acquisition, Investigation; Roland Brosch, Brigitte Gicquel, Supervision, Funding acquisition; Ludovic Tailleux, Conceptualization, Formal analysis, Validation, Investigation, Visualization, Methodology, Supervision, Funding acquisition, Writing - original draft, Project administration, Writing - review and editing

## Author ORCIDs

Maximiliano G Gutierrez (ID) http://orcid.org/0000-0003-3199-0337
Ludovic Tailleux (ID) https://orcid.org/0000-0003-3931-1052

## Ethics

Human subjects: Buffy coats were obtained from healthy donors after informed consent. The blood collection protocols were approved by both the French Ministry of Research and a French Ethics Committee. The blood collection was carried out in accordance with these approved protocols by the Etablissement Français du Sang (EFS, number 12/EFS/134).

## Decision letter and Author response

Decision letter https://doi.org/10.7554/eLife.55692.sa1
Author response https://doi.org/10.7554/eLife.55692.sa2

# Additional files

## Supplementary files

• Supplementary file 1. Supplementary materials and methods.
• Supplementary file 2. Oligonucleotide sequences.
• Transparent reporting form

## Data availability

The raw fastq files of BDQ-treated cells have been deposited in NCBI's Gene Expression Omnibus (Edgar et al., 2002) and are accessible through GEO Series accession number GSE133145. The raw fastq files of cells stimulated with heat-killed MTB or treated with different antibiotics are accessible through GEO Series accession numbers GSE143627 and GSE143731.

The following datasets were generated:

| Author(s) | Year | Dataset title | Dataset URL | Database and Identifier |
|---|---|---|---|---|
| Giraud-Gatineau A, Tailleux L | 2019 | Bedaquiline remodels the macrophage response | https://www.ncbi.nlm. nih.gov/geo/query/acc. cgi?acc=GSE133145 | NCBI Gene Expression Omnibus, GSE133145 |
| Giraud-Gatineau A, Tailleux L | 2020 | Inactivated *M. tuberculosis* and *M. tuberculosis* Infection remodels the macrophage response | https://www.ncbi.nlm. nih.gov/geo/query/acc. cgi?acc=GSE143627 | NCBI Gene Expression Omnibus, GSE143627 |
| Giraud-Gatineau A, Tailleux L | 2020 | Genome-wide gene expression profiling of anti-tuberculosis drugs- | https://www.ncbi.nlm. nih.gov/geo/query/acc. | NCBI Gene Expression Omnibus, |

| treated macrophages | cgi?acc=GSE143731 | GSE143731 |
| --- | --- | --- |

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
