## [Decision Letter]

**Acceptance summary:**

This paper provides evidence that the antibiotic bedaquiline not only has direct anti-microbial effects but may also act on host cells to enhance their capacity for killing bacteria. Bedaquiline is an important antibiotic for the treatment of tuberculosis, and a better understanding of its activities is important for developing next-generation therapeutics for this globally significant disease. More-broadly, understanding the host-directed effects of antibiotics is also significant for other bacterial infections for which drug resistance is increasing.

**Decision letter after peer review:**

[Editors’ note: the authors submitted for reconsideration following the decision after peer review. What follows is the decision letter after the first round of review.]

Thank you for submitting your work entitled "The antibiotic bedaquiline activates host macrophage innate immune resistance to bacterial infection" for consideration by *eLife*. Your article has been reviewed by three peer reviewers, and the evaluation has been overseen by a Reviewing Editor and a Senior Editor.

Our decision has been reached after consultation between the reviewers. We found that the work addresses an important topic and is interesting. However, the reviewers have raised substantial technical concerns. Therefore, the decision is to reject your manuscript without requesting a revision. Nevertheless, in addition to the individual reviews, we are providing a consolidated review (below) that captures the consensus of the three reviewers in case you feel that the concerns are easily addressable within a reasonable timeframe.

1) A major concern is that the manuscript does not cleanly demonstrate that the effects of BDQ observed on MTB growing in cells is in fact due to the effects of BDQ on the host cells vs. effects of BDQ on MTB. The use of a BDQ resistant mutant is a nice attempt to address this concern, but it is not entirely satisfactory. First, the data shown in Figure 1—figure supplement 2C demonstrate that BDQ (5µg/mL) does appear to have an effect on growth of the resistant mutant. Second, even if growth of the bacteria is unimpeded in broth, the antibiotic may still have an effect on the bacteria in cells. Even a minor effect on MTB (e.g., lysis of a few bacteria) might have a large effect on the host response. Unless this central concern can be satisfactorily addressed (which we fear it may not be) then we do not feel the paper would be a strong candidate for *eLife*.

2) Another major concern relates to the presentation of the RNAseq data in Figure 1A-B and Figure supplement 3A-B. Figure 1AB shows the effect of BDQ on MTB-infected macrophages, whereas Figure supplement 3A-B shows the effect of BDQ on uninfected macrophages. If BDQ acts on host cells independent of an effect on bacteria, then we would expect similar genes to be induced in both conditions. However, the data are presented only at the level of GO-term annotations. We require gene-level expression data comparing gene induction in all four conditions (+/- BDQ and +/- MTB) in a single analysis so that we can assess whether in fact the effects of BDQ are similar +/- MTB. It may well be that MTB modulates the host response to BDQ, e.g., via activation of TLRs. In such a case, it might be useful to compare the response of cells infected with live TB +/- BDQ with the response to cells treated with TLR agonists or killed bacteria +/- BDQ. If, however, the main effects of BDQ on host gene expression only appear to occur in conjunction with infection with live MTB, then we would be concerned that the effects of BDQ are via their effects on MTB and not direct host-directed effects.

3) Another major concern is that there is no direct evidence that the apparent effects of BDQ on MTB growing in cells is due to induction of TFEB. This should be shown with TFEB mutant cells. These cells could also be used to show that the induction of lysosomal-related genes and increased lysotracker staining by BDQ is due to TFEB activation.

4) As raised by reviewer 2, it is imperative to show that the synergy between BDQ and PZA on the BDQ-resistant mutant does not occur in broth.

Other comments:

- A MIC for BDQ on the resistant mutant should be reported.

- Figure 1—figure supplement 2B is missing information about what concentration of BDQ was used.

- Reviewer 2 raised a number of questions about the data in Figure 1—figure supplement 2C that should be addressed.

- As raised by reviewer 3, the proposed mechanism connecting BDQ to calcium and TFEB activation is only weakly supported by data. The authors should consider toning down their conclusions related to this part of the paper.

Reviewer #1:

Tuberculosis remains a major global problem and kills more people than any other single pathogen. This paper addresses the mechanism of action of one of the most important new tuberculosis drugs, bedaquiline (BDQ), and shows that in addition to its well-known direct anti-microbial effects, BDQ also has significant 'host-directed' effects that can contribute to its anti-microbial activity. In particular, BDQ is shown to enhance lysosomal activity in macrophages. This appears to be via activation of the host transcription factor, TFEB.

In general the paper is well-written, and the experiments are well-described and well-performed. There is significant current interest in the idea of host-directed therapies for TB, and the idea that a major new TB drug might act in part via its effects on host cells is interesting and potentially very important. I have only a few minor comments:

1) Transcriptional profiling is performed on both MTB-infected and in uninfected cells. It is claimed that the effects of BDQ are "similar" in both situations. However, the way the data are presented makes it hard to see how similar or different the effects of BDQ are +/- MTB. Could the authors present a heat-map (indicating key genes, particularly the lysosomal signature) with all four conditions (+/- MTB, +/- BDQ) in a single analysis/panel? If there are significant differences, it might be relevant to comment on these.

2) A weakness of the manuscript is that the direct host-cell target of BDQ is not identified. Thus, the mechanism leading to TFEB induction is unclear. However, I tend to feel that identification of the target of BDQ is beyond what could be reasonably asked of this study. However, perhaps the authors could at least briefly discuss what the host cell target of BDQ might be and acknowledge that they have not identified it. In addition, the evidence that the effects of BDQ require TFEB is only indirect. It would thus be desirable, if feasible, for the authors to show, e.g., with TFEB knock out cells, that the effects of BDQ on lysosome-related gene induction does require TFEB.

Reviewer #2:

The manuscript by Giraud-Gatineau et al. concludes that the newest anti-tuberculosis antibiotic, bedaquiline (BDQ), works by not only inhibiting ATP synthase in the bacterium, but also by synergistically working directly on host macrophages to boost their antimicrobial effects during infection. There have been many reports of similar kinds of effects of different antibiotic regimens on activating immune responses the host, but nearly all suffer from the inherent difficulty in conclusively determining whether the "host effects" of the antibiotic are due to direct action on macrophage pathways or whether the effects on the host are secondary to the effects of the antibiotic on the physiology of the bacteria. Indeed, each of the papers cited by the authors on this subject suffer from this problem, and the field remains murky.

It would be an exciting finding that BDQ activates host macrophage pathways (perhaps by partial inhibition of the host ATP synthase?), but unfortunately my main criticism of this paper is that I find that there are multiple issues that center around the main problem of whether BDQ actually increases the antibacterial capacity of macrophages by directly activating host pathways. I certainly appreciate the difficulty of the task, and the authors consider bacterial load as one potential factor by using a BDQ-resistant mutant, but many of my concerns relate to the data itself of these key experiments, as well as the possibility of effects on bacterial physiology induced by the antibiotic. For example, even a small amount of lysis of the bacteria could influence host transcriptional responses. Likewise, it seems that comparing resistance of a bacterium in culture vs. in a cell doesn't necessarily show that the effect is on the host. It seems likely that the antibiotic is still working solely on the bug, but the host environment sensitizes the bacterium to the antibiotic.

1) It seems likely to me that the transcriptional (Figure 1) and functional effects (Figure 2) of BDQ on infected macrophages is due to indirect effects on bacterial physiology. The use of the BDQ-resistant mutant is an important experiment, but I have doubts that the concentration used for the macrophage infection studies (5µg/mL) does not have any effect on the bacteria. This concentration apparently inhibits the growth of the resistant mutant Figure 1—figure supplement 2C inside macrophages, and this could be due to either activation of antimicrobial host effects or partial inhibition of bacterial growth. It is not clear what effects this concentration has on the bacteria, the mutant was selected on 0.3 µg/mL BDQ, and I could not find an MIC stated in the manuscript. The growth curve of bacteria grown in axenic culture +/- BDQ (panel B) is missing any information about the concentration of BDQ used, and there is no direct comparison of the growth kinetics between wild-type and mutant bacteria at 5 µg/mL. Moreover, the assumption is that treatment with 5µg/mL of BDQ doesn't change the bacteria at all, and thus any differences must be due to host modulation, yet I think there are multiple reasons to doubt this.

2) There are a number of other worrisome details about the experiment displayed in Figure 1—figure supplement 2C. First, it is curious that neither the wild-type nor resistant bacteria grew very well between the 18hour and 5day time points in the absence of BDQ. It appears they double at best, but much more would be expected. Second, the BDQ-resistant mutant didn't grow at all over this 4+ day time course in the presence of 5µg/mL BDQ. It seems unlikely to me that this effects of BDQ are solely due to host activation. Third, why are the CFU levels of the mutant bacteria so much higher than the wild-type at 18 hours?

3) It seems important to demonstrate in all of their bacterial infections, including Figure 4, that the macrophage monolayers are intact and of equal density amongst all the treatments. Indeed, while the authors test the effects of BDQ on macrophage viability in the absence of bacteria, it seems important to assess its effect during the heightened inflammatory response during infection.

4) I believe that the results with the host effects of BDQ would be more conclusive if they could show that treatment with different antibiotics gives rise to different results.

5) The authors also perform treatment of macrophages with BDQ in the absence of infection, and claim that similar pathways are activated, inferring that these effects are independent of the presence of bacteria. Unfortunately, these comparisons were evaluated at the level of GO pathway annotation, and I believe strongly that if BDQ is directly activating gene expression, perhaps via PTFB, that the comparison should be performed at the gene level, i.e. as done in Figure 2A.

6) I find the conclusion drawn from the apparent synergy btw PZA and BDQ is misleading for two reasons. First of all, given the residual sensitivity of the resistant mutant to high levels of BDQ, it seems imperative that this synergy be tested in culture. Second, given the model that BDQ activates host antimicrobial activities, I expect that BDQ would also synergize with other antibiotics besides PZA.

7) Much of the lysosome results in Figure 2 is complicated by the mild sensitivity of the resistant strain to BDQ. The one experiment that is interesting is the no-infection in panel G, but there is only a correlation, there is no evidence provided that this is the host effect of BDQ.

Reviewer #3:

In this manuscript, Giraud-Gatineau et al. describe how bedaquiline (BDQ), an antibiotic in clinical use for TB, activates bactericidal functions in human primary macrophages. Transcriptional analysis of such treated macrophages leads them to examine the autophagy-lysosomal pathway, which they find is upregulated and necessary for the effect of BDQ in reducing *M. tuberculosis* load in infected macrophages in vitro. Finally, they show that transcription factor TFEB is activated by BDQ, and that the activity of BDQ in macrophages is dependent on calcium, suggesting that the MCOLN1-calcineurin-TFEB pathway described by Medina et al., (2015) is implicated.

Overall, the concept of antibiotics serving dual functions as host-directed therapies is appealing and of great interest. However, there are several major concerns that must be addressed:

1) The proposed mechanism of enhancement of bactericidal activity is through TFEB, but the only evidence presented is correlative (TFEB immunofluorescence showing TFEB in the nucleus). This must be directly tested using loss-of-function approaches.

2) Similarly, the implication from the RNAseq, LysoTracker, and TFEB IF is that TFEB upregulates the genes responsible for lysosomal biogenesis and autophagy, thus increasing bacterial clearance. The inducible recruitment of TFEB to relevant promoters must be directly tested.

3) The mechanism connecting BDQ to calcium release and TFEB activation is not shown. Neither is the requirement for calcineurin.

4) Although the synergistic effect of BDQ with PZA seems to approach 1 log (~10% survival) killing of *M. tuberculosis*, the effect size of BDQ alone (Figure 3F) is quite small. This raises questions of how significant these observations are to TB. This is important, since the whole setup is to find new therapeutic agents to control MDR TB.

5) The authors use BAPTA (extracellular) to eliminate calcium and test the effect of loss of TFEB activation on bacterial CFU. This is done with *S. aureus* (not TB). Also, if they really are using BAPTA, they would not be inhibiting intracellular calcium release but import from extracellular stores, changing the interpretation of the experiment.

[Editors’ note: further revisions were suggested prior to acceptance, as described below.]

Thank you for resubmitting your work entitled "The antibiotic bedaquiline activates host macrophage innate immune resistance to bacterial infection" for further consideration by *eLife*. Your article has been reviewed by three peer reviewers, and the evaluation has been overseen by a Reviewing Editor and a Senior Editor.

The manuscript has been improved and the reviewers are generally enthusiastic about publication but there are some remaining issues that need to be addressed before acceptance. Please revise your manuscript to address these issues.

*Reviewer #1:*

The revised manuscript addresses the main concerns of this reviewer. Overall, I think the data strongly indicate that the antibiotic bedaquiline (BDQ) can induce antibacterial responses by direct action on host cells. The mechanism of action of BDQ is of great importance to the TB field, and the concept that antibiotics can act (in part) by targeting the host is of even broader importance.

My only query relates to new data (Figure 7) showing that human macrophages induce nitrite, which the authors claim is derived from host NO. As far as I am aware, there has been considerable difficulty demonstrating induction of NO by human macrophages in culture. In some cases, the NO2- observed has actually been produced by the bacteria themselves (see Bussel, Zhang and Nathan, 2013). The authors may want to take this into account.

---

## [Author Response]

[Editors’ note: the authors resubmitted a revised version of the paper for consideration. What follows is the authors’ response to the first round of review.]

[…]1) A major concern is that the manuscript does not cleanly demonstrate that the effects of BDQ observed on MTB growing in cells is in fact due to the effects of BDQ on the host cells vs. effects of BDQ on MTB. The use of a BDQ resistant mutant is a nice attempt to address this concern, but it is not entirely satisfactory. First, the data shown in Figure 1—figure supplement 2C demonstrate that BDQ (5µg/mL) does appear to have an effect on growth of the resistant mutant. Second, even if growth of the bacteria is unimpeded in broth, the antibiotic may still have an effect on the bacteria in cells. Even a minor effect on MTB (e.g., lysis of a few bacteria) might have a large effect on the host response. Unless this central concern can be satisfactorily addressed (which we fear it may not be) then we do not feel the paper would be a strong candidate for eLife.

We thank the reviewer 2 for raising this point, acknowledging that this may be due to problems in our writing. Our main goal was to show that BDQ can activate macrophages, making them resistant to any bacterial infections, and not just to infection with MTB. In the revised manuscript, we present new data, which clearly show that BDQ induces bactericidal functions by directly targeting the host.

The first question is to what extent is the MTB mutant resistant to BDQ (BDQr-MTB). The MIC_99_ of the mutant is 36 µg/mL, which is much higher than the concentration that we used. In our paper, BDQ is used at a maximum concentration of 5 µg/mL during 48 hours and 1 µg/mL during 5 days. At these concentrations, we never observed bacterial killing nor decrease of bacterial growth in liquid medium (Figure 1—figure supplement 2B). We also showed that the bacillary load of resistant MTB inside macrophages is the same in untreated cells as in cells after 18 hours of BDQ treatment (the time point chosen for the RNAseq experiment). In contrast, in the same experiment using BDQ-susceptible MTB there is a 70% decrease in the bacillary load (Figure 1—figure supplement 2C).

We agree that BDQ might kill a few intracellular bacteria (even if we did not detect any changes by CFUs), but it is very unlikely that this effect could explain the macrophage resistance to bacterial infection. First, the effect of BDQ can also be inhibited by 3-MA and BAF, whereas these two autophagy inhibitors have no effect on untreated cells (Figure 6F). These results suggest a host-targeted effect of BDQ. The regulation of the lysosomal pathway by BDQ is also observed in uninfected macrophages and in cells stimulated with heat-inactivated MTB or with TLR agonists such as LPS and Pam3CSK4 (Figure 3—figure supplement 2). Such changes were not observed in cells treated with the first line anti-TB drugs ethambutol (EMB), isoniazid (INH), pyrazinamide (PZA) and rifampicin (RIF) (new transcriptome analysis presented in Figure 5A-C and Figure 5—figure supplement 2, and LysoTracker staining analyzed by flow cytometry Figure 5D).

If the lysis of a few bacteria by BDQ induced macrophage activation, then we would expect a greater response when the cells are stimulated by heat-killed bacteria or drug-susceptible MTB than with BDQr-MTB. Again, as shown in Figure 3—figure supplement 2, we did not observe any differences in the intensity of LysoTracker staining. Similar results have also been obtained when comparing the expression level of lysosomal genes. Overall, our results strongly argue for a host-dependent effect of BDQ.

2) Another major concern relates to the presentation of the RNAseq data in Figure 1A-B and Figure supplement 3A-B. Figure 1AB shows the effect of BDQ on MTB-infected macrophages, whereas Figure supplement 3A-B shows the effect of BDQ on uninfected macrophages. If BDQ acts on host cells independent of an effect on bacteria, then we would expect similar genes to be induced in both conditions. However, the data are presented only at the level of GO-term annotations. We require gene-level expression data comparing gene induction in all four conditions (+/- BDQ and +/- MTB) in a single analysis so that we can assess whether in fact the effects of BDQ are similar +/- MTB. It may well be that MTB modulates the host response to BDQ, e.g., via activation of TLRs. In such a case, it might be useful to compare the response of cells infected with live TB +/- BDQ with the response to cells treated with TLR agonists or killed bacteria +/- BDQ. If, however, the main effects of BDQ on host gene expression only appear to occur in conjunction with infection with live MTB, then we would be concerned that the effects of BDQ are via their effects on MTB and not direct host-directed effects.

Here we present new analysis, comparing gene expression level in all four conditions (+/- BDQ and +/- MTB). Upon BDQ treatment, 127 and 1043 genes, respectively, were differentially expressed in naïve cells and in MTB-infected macrophages, and the expression of 452 genes was modulated in both conditions. It is not surprising that the expression of more genes was affected in MTB-infected cells as MTB infection induces extensive remodeling of the transcriptome (Barreiro et al., 2012; Tailleux et al., 2008). We then classified all these genes by performing gene-set enrichment analysis. We showed that the expression of 70% of the lysosomal gene differentially expressed in MTB-infected macrophages, were also up-regulated in naïve cells upon BDQ treatment. These results are now included in Figure 1D and Figure 3A. We also performed additional experiments to determine whether the effects of BDQ on host gene expression were dependent of infection with live MTB. Briefly, cells were untreated or stimulated with LPS (TLR4 agonist), Pam3CSK4 (TLR1/2 agonist), heat-killed bacteria, drug-susceptible MTB or BDQr-MTB, and treated with BDQ. After 18 hours, RNA was collected and we performed RT-qPCR on a panel of lysosomal genes. We also analyzed the intensity of the LysoTracker staining using flow cytometry (*Figure 3—figure supplement 2*). Our results clearly show that the main effects on lysosome biogenesis/activation occurred with BDQ treatment and were not exclusively seen after infection with live MTB.

3) Another major concern is that there is no direct evidence that the apparent effects of BDQ on MTB growing in cells is due to induction of TFEB. This should be shown with TFEB mutant cells. These cells could also be used to show that the induction of lysosomal-related genes and increased lysotracker staining by BDQ is due to TFEB activation.

Using siRNA-mediated gene silencing (Troegeler et al., 2014), we inactivated the expression of TFEB in macrophages and confirmed its major role in the resistance to bacterial infection upon BDQ treatment. We also present a chromatin immunoprecipitation assay with anti-TFEB antibody. As suggested by reviewers 1 and 3, we showed that TFEB activation is required for (i) the increased LysoTracker staining (Figure 9G), (ii) the induction of a panel of lysosomal-related genes (Figure 9H), and (iii) a better control of *S. aureus* and MTB infection upon BDQ treatment (Figure 9I-J).

4) As raised by reviewer 2, it is imperative to show that the synergy between BDQ and PZA on the BDQ-resistant mutant does not occur in broth.

We performed the suggested experiment. As shown in Figure 4B, we found no synergy between BDQ and PZA on the BDQ-resistant mutant cultivated in Middlebrook 7H9 liquid medium.

- A MIC for BDQ on the resistant mutant should be reported.

This has been added in the revised version. We determined the MIC_99_ (defined as the concentration required to prevent 99% growth) of both wild-type and BDQ-resistant MTB by the colorimetric resazurin microtiter assay (Palomino et al., 2002). The MIC for susceptible MTB was 0.07 µg/mL, a value similar to previously published studies (Andries et al., 2005), and the MIC_99_ of the BDQr-MTB was 36 µg/mL.

- Figure 1—figure supplement 2B is missing information about what concentration of BDQ was used.

This has been corrected in the revised version.

- Reviewer 2 raised a number of questions about the data in Figure 1—figure supplement 2C that should be addressed.

We thank the reviewer for the points raised about the experiment displayed in Figure 1—figure supplement 2C, which could question the virulence of the both wild-type and BDQ-resistant MTB. In the former version of our paper, we only quantified the number of intracellular bacteria, excluding the extracellular bacilli released from necrotic cells. This gave the wrong impression that bacteria do not multiply within macrophages. In the revised manuscript we provide new data from CFUs and confocal microscopy showing that wild-type and BDQ-resistant MTB are both virulent and replicate inside macrophages (Figure supplement 2C-E).

- As raised by reviewer 3, the proposed mechanism connecting BDQ to calcium and TFEB activation is only weakly supported by data. The authors should consider toning down their conclusions related to this part of the paper.

The calcium chelation experiments were performed with BAPTA-AM, not BAPTA (please see our comments to reviewer #3). We would like to apologize for this error and have corrected it in the revised manuscript.

Reviewer #1:[…]1) Transcriptional profiling is performed on both MTB-infected and in uninfected cells. It is claimed that the effects of BDQ are "similar" in both situations. However, the way the data are presented makes it hard to see how similar or different the effects of BDQ are +/- MTB. Could the authors present a heat-map (indicating key genes, particularly the lysosomal signature) with all four conditions (+/- MTB, +/- BDQ) in a single analysis/panel? If there are significant differences, it might be relevant to comment on these.

We thank the reviewer for his positive feedback and for fair suggestions. In the revised manuscript, we present a new analysis (Figure 1). MTB infection induces important transcriptome remodeling (7,093 genes were differentially expressed upon infection with BDQr-MTB) so it is not surprising that the expression of more genes were altered after BDQ treatment of MTB-infected macrophages than uninfected macrophages. In the new version, we comment on these and present functional annotation clustering of the corresponding genes (Figure 1A-B). In the former version of our manuscript, we wrote that “We observed similar results with uninfected Mφs-treated with BDQ […] indicating that the effect of BDQ is not dependent on MTB infection”. We were principally referring here to the lysosome pathway and lipid metabolism. 452 genes were differentially expressed in both uninfected- and MTB infected cells. Classification of these genes using gene-set enrichment analysis shows an enrichment for genes associated with lysosome (Figure 1C-D). The expression of 70% of the lysosomal genes found to be differentially expressed in BDQr-MTB-infected cells, were also regulated in naïve cells upon BDQ treatment (Figure 3A). As suggested, we have added a first heat-map of the genes most up- and down-regulated in naïve- and MTB-infected cells treated with BDQ (Figure 1D), and a second heat-map focusing on the expression of lysosomal genes in all four conditions +/- MTB +/- BDQ (Figure 3A).

2) A weakness of the manuscript is that the direct host-cell target of BDQ is not identified. Thus, the mechanism leading to TFEB induction is unclear. However, I tend to feel that identification of the target of BDQ is beyond what could be reasonably asked of this study. However, perhaps the authors could at least briefly discuss what the host cell target of BDQ might be and acknowledge that they have not identified it. In addition, the evidence that the effects of BDQ require TFEB is only indirect. It would thus be desirable, if feasible, for the authors to show, e.g., with TFEB knock out cells, that the effects of BDQ on lysosome-related gene induction does require TFEB.

Finding the direct host-cell target of BDQ would have been a major breakthrough, but unfortunately, we were unable to identify the target. However, it is possible that BDQ could modulate the host response by interfering with ion homeostasis inside the lysosome. Greenwood et al., recently showed that BDQ accumulated primarily in host cell lipid droplets (Greenwood et al., 2019). It is tempting to speculate that these droplets are actually lysosomes. Due to its high hydrophobicity, BDQ might accumulate in lysosomal membranes and thereby change the transmembrane ion permeability. In agreement with this hypothesis, it has been shown that BDQ can accumulate at the lipid membrane of liposomes and act as a H^+^/K^+^ ionophore (Hards et al., 2018). The ensuing lysosomal stress could then facilitate the dissociation of mTOR from the lysosomal membrane (Plescher et al., 2015) and the activation of TFEB. Interestingly, BDQ increases the expression of mucolipin 1 gene (*mcoln1*, Figure 3A and Figure 3—figure supplement 2A). Recently, Medina et al. showed that lysosomal Ca^2+^ is released through MCOLN1 and activates calcineurin, which binds and dephosphorylates TFEB (Medina et al., 2015) (Medina et al., 2015). These two hypotheses are now mentioned in the Discussion section of our paper.

In the revised manuscript, we present additional experiments, including silencing of TFEB expression using siRNA and chromatin purification with anti-TFEB antibody. The results show that the effects of BDQ on the induction of genes linked to lysosomes indeed require TFEB (Figure 9 and Figure 9—figure supplement 1).

Reviewer #2:[…]1) It seems likely to me that the transcriptional (Figure 1) and functional effects (Figures 2- of BDQ on infected macrophages is due to indirect effects on bacterial physiology. The use of the BDQ-resistant mutant is an important experiment, but I have doubts that the concentration used for the macrophage infection studies (5µg/mL) does not have any effect on the bacteria. This concentration apparently inhibits the growth of the resistant mutant Figure S1C inside macrophages, and this could be due to either activation of antimicrobial host effects or partial inhibition of bacterial growth. It is not clear what effects this concentration has on the bacteria, the mutant was selected on 0.3 µg/mL BDQ, and I could not find an MIC stated in the manuscript. The growth curve of bacteria grown in axenic culture +/- BDQ (panel B) is missing any information about the concentration of BDQ used, and there is no direct comparison of the growth kinetics between wild-type and mutant bacteria at 5 µg/mL. Moreover, the assumption is that treatment with 5µg/mL of BDQ doesn't change the bacteria at all, and thus any differences must be due to host modulation, yet I think there are multiple reasons to doubt this.

We thank the reviewer 2 for raising the question to what extent is the MTB mutant resistant to BDQ (BDQr-MTB). We first determined the MIC_99_ (defined as the concentration required to prevent 99% growth) of both wild-type- and BDQ-resistant MTB by the colorimetric resazurin microtiter assay (Palomino et al., 2002). The MIC for susceptible MTB was 0.07 µg/mL, a value similar to previously published study (Andries et al., 2005), while the MIC_99_ of the BDQr-MTB was 36 µg/mL. The CFUs and the growth curves of both WT and BDQr-MTB, with and without different concentrations of BDQ are now included in the manuscript (Figure 1—figure supplement 2B-C). Unlike drug-susceptible MTB, BDQrMTB continues to growth in the presence of 5 µg/mL BDQ, and we observed a slight decrease in growth only after 7 days of treatment. In our paper, BDQ is used at a maximum concentration of 5 µg/mL during 48 h and 1 µg/mL during 5 days. At these concentrations, we observed neither bacterial killing nor a decrease of bacterial growth in liquid medium (Figure 1—Figure supplement 2B-C). We also show that the bacillary load of resistant MTB inside macrophages is the same in untreated cells as in cells after 18 h of BDQ treatment (the time point chosen for the RNAseq experiment). In contrast, in the same experiment using BDQ-susceptible MTB there is a 70% decrease in the bacillary load (Figure 1—figure supplement 2C).

We agree that we cannot exclude the possibility that BDQ may affect the metabolism of BDQr-MTB, but it seems unlikely that this would explain the macrophage resistance to bacterial infection upon BDQ treatment. First, the effect of BDQ can be inhibited by 3-MA and BAF, whereas these two autophagy inhibitors have no effect on untreated cells (Figure 6F). These results suggest a host-targeted effect of BDQ. The regulation of the lysosomal pathway by BDQ seen in MTB infected macrophages was also observed in naïve macrophages and in cells stimulated with heat-inactivated MTB or with TLR agonists LPS and Pam3CSK4 (Figure 3—figure supplement 2). These changes were not observed in cells treated with the first line anti-TB drugs ethambutol (EMB), isoniazid (INH), pyrazinamide (PZA) and rifampin (RIF) (new transcriptome analysis presented in Figure 5A-B-C and Figure 6—figure supplement 2, and LysoTracker staining analyzed by flow cytometry Figure 5D).

If the lysis of a few bacteria by BDQ induces macrophage activation, we would expect a greater response when the cells are stimulated by heat-killed bacteria or drug-susceptible MTB than was observed with BDQr-MTB. Again, as shown in Figure 3—figure supplement 2B, we did not observe any differences in the intensity of LysoTracker staining. Similar results have also been obtained when comparing the expression level of lysosomal genes.

Overall, our results strongly argue for a host-dependent effect of BDQ.

2) There are a number of other worrisome details about the experiment displayed in Figure S1C. First, it is curious that neither the wild-type nor resistant bacteria grew very well between the 18hour and 5day time points in the absence of BDQ. It appears they double at best, but much more would be expected. Second, the BDQ-resistant mutant didn't grow at all over this 4+ day time course in the presence of 5µg/mL BDQ. It seems unlikely to me that this effects of BDQ are solely due to host activation. Third, why are the CFU levels of the mutant bacteria so much higher than the wild-type at 18hours?

We thank the reviewer for the points raised about the experiment displayed in the former Figure 1—figure supplement 1C, which could question the virulence of both wild-type and BDQ-resistant MTB. In the former version of our paper, we only quantified the number of intracellular bacteria, excluding the extracellular bacillus released from necrotic cells. This gave the erroneous impression that the bacteria did not multiply inside macrophages. In the revised manuscript, we provide new data from CFUs and confocal microscopy showing that wild-type and BDQ-resistant MTB are virulent and replicate within macrophages.

3) It seems important to demonstrate in all of their bacterial infections, including Figure 4, that the macrophage monolayers are intact and of equal density amongst all the treatments. Indeed, while the authors test the effects of BDQ on macrophage viability in the absence of bacteria, it seems important to assess its effect during the heightened inflammatory response during infection.

The viability of naïve and infected macrophages, treated with the different molecules used in this study, were assessed using the MTT or LDH assay. As shown in Figure 1—figure supplement 1, we noted no toxicity of the compounds. As expected, we only observe a decrease in cell viability of MTB-infected macrophages after 48 h of infection.

4) I believe that the results with the host effects of BDQ would be more conclusive if they could show that treatment with different antibiotics gives rise to different results.

In the initial version of our manuscript, we only tested whether first-line anti-TB drugs could induce an increase in LysoTracker staining. We agree that further analysis of the effects of other drugs on the macrophages response would be more conclusive. We therefore used RNAseq to characterize the genome-wide gene expression profiles of naïve cells and cells stimulated with heat-killed MTB, and treated with ethambutol (EMB), isoniazid (INH), pyrazinamide (PZA) or rifampicin (RIF). We chose heat-killed MTB instead of drug-resistant MTB for the following reasons:

(i) We cannot work with multidrug-resistant MTB in our BSL3 facility, and the generation of resistant mutants for each antibiotic would not have been reasonable in terms of time to resubmit a revised version of our manuscript.

(ii) Most of the genes upon that were differentially expressed after MTB infection are also modulated upon stimulation with heat-killed MTB ((Pacis et al., 2015) and Figure 6—figure supplement 1).

(iii) The activation of the lysosomal pathway by BDQ does not require infection with live MTB, as most of the lysosomal genes are also differentially expressed in naïve cells exposed to BDQ (Figure 3A).

Following treatment, only RIF and PZA significantly modulate gene expression in macrophages. 556 and 752 genes were differentially expressed in cells stimulated with heat-killed bacteria and exposed to PZA and RIF respectively (Figure 5A). We classified these genes by performing gene-set enrichment analysis and confirmed that the genes in the lysosomal pathway were not induced upon RIF or PZA treatment. The expression of only 2 genes belonging to this pathway was upregulated by RIF, and only one by PZA, compared to 46 whose expression was modulated by BDQ (FDR<5%, absLogFC>0.1, Figure 5B-C and Figure 5—figure supplement 2).

5) The authors also perform treatment of macrophages with BDQ in the absence of infection, and claim that similar pathways are activated, inferring that these effects are independent of the presence of bacteria. Unfortunately, these comparisons were evaluated at the level of GO pathway annotation, and I believe strongly that if BDQ is directly activating gene expression, perhaps via PTFB, that the comparison should be performed at the gene level, i.e. as done in Figure 2A.

In the revised manuscript, we present new data comparing the gene expression level in all four conditions (+/- BDQ and +/- MTB), and as suggested by reviewer 2, we show a heat-map with the differentially expressed genes associated with the lysosome pathway (Figure 3A). The expression of 70% of the lysosomal pathway genes, differentially expressed in MTB-infected macrophage, were also upregulated in naïve cells upon BDQ treatment.

We also performed additional experiments to demonstrate that the effects of BDQ on host gene expression were independent of infection with live MTB. Briefly, cells were untreated or stimulated with LPS (TLR4 agonist), Pam3CSK4 (TLR1/2 agonist), heat-killed bacteria, drug-susceptible MTB or BDQr-MTB, and treated with BDQ. After 18 h, RNA was collected, and RT-qPCR performed on a panel of lysosomal genes. We also analyzed the intensity of the LysoTracker staining using flow cytometry (Figure 3—figure supplement 1). The results clearly show that the effect of BDQ on lysosome biogenesis/activation is independent of MTB infection and of Toll-like receptor stimulation.

6) I find the conclusion drawn from the apparent synergy btw PZA and BDQ is misleading for two reasons. First of all, given the residual sensitivity of the resistant mutant to high levels of BDQ, it seems imperative that this synergy be tested in culture. Second, given the model that BDQ activates host antimicrobial activities, I expect that BDQ would also synergize with other antibiotics besides PZA.

We tested the synergy between BDQ and PZA in culture liquid medium using increasing concentrations of PZA (from 0 to 400 µg/mL) with or without BDQ (1 µg/mL). At this BDQ concentration, the growth of BDQr-MTB is not affected by BDQ over a period of 12 days (Figure 1—figure supplement 2B) and we observed an increase of the DQ-BSA staining in macrophages (Figure 3F). We observed no synergy between BDQ and PZA in culture liquid medium, suggesting a hostdependent mechanism. We next tested whether BDQ would synergize with the other first-line anti-TB drugs in liquid culture or in BDQr-MTB-infected macrophages and found that BDQ did not potentiate the activity of INH, RIF and EMB in either case. While we cannot exclude the possibility that BDQ may have additive effects with other anti-TB drugs, as has been described (Ibrahim et al., 2007), we saw no evidence of synergism with any of the first-line agents (Figure 4—figure supplement 1).

7) Much of the lysosome results in Figure 2 is complicated by the mild sensitivity of the resistant strain to BDQ. The one experiment that is interesting is the no-infection in panel G, but there is only a correlation, there is no evidence provided that this is the host effect of BDQ.

In the revised version, we add a new supplemental figure describing the lysosomal regulation in uninfected cells (Figure 3—figure supplement 1). As we mention in the text, lysosome biogenesis/activation is increased in both naïve and BDQr-MTB-infected macrophages upon BDQ treatment.

Reviewer #3:[…]1) The proposed mechanism of enhancement of bactericidal activity is through TFEB, but the only evidence presented is correlative (TFEB immunofluorescence showing TFEB in the nucleus). This must be directly tested using loss-of-function approaches.

To study the role of TFEB in the enhancement of bactericidal activity upon BDQ treatment in more depth, we inactivated TFEB expression in human monocyte-derived macrophages using siRNA-mediated gene silencing (Troegeler et al., 2014). After treating the cells with BDQ, we performed RTqPCR on a panel of lysosomal genes and analyzed the intensity of the LysoTracker staining using flow cytometry. In TFEB-silenced cells, BDQ does not increase LysoTracker staining (Figure 9G), nor the expression of *trpm2*, *scarb1*, *atp6v0a1* or *mcoln1* (Figure 9H). We also infected macrophages with *S. aureus* and MTB and showed that TFEB activation is required for better control of both pathogens upon BDQ treatment (Figure 9I-J). In addition, we present a chromatin immunoprecipitation assay using an anti-TFEB antibody (please see the comment below).

2) Similarly, the implication from the RNAseq, LysoTracker, and TFEB IF is that TFEB upregulates the genes responsible for lysosomal biogenesis and autophagy, thus increasing bacterial clearance. The inducible recruitment of TFEB to relevant promoters must be directly tested.

We evaluated the recruitment of TFEB to the promoters of some lysosomal genes. Briefly, naïve- and heat-killed-stimulated cells treated with BDQ were lysed, and chromatin was harvested and fragmented using sonication. The chromatin was then subjected to immunoprecipitation using an antiTFEB antibody. The isolated DNA was then analyzed by RT-qPCR with primers specific for *gla* and *mcoln1* promoters. As expected, BDQ induces the recruitment of TFEB to these promoters (Figure 8—figure supplement 1).

3) The mechanism connecting BDQ to calcium release and TFEB activation is not shown. Neither is the requirement for calcineurin.

Unfortunately, we were unable to identify the mechanism connecting BDQ to TFEB activation.

Greenwood et al., recently showed that BDQ accumulated primarily in host cell lipid droplets (Greenwood et al., 2019). It is tempting to speculate that these droplets are actually lysosomes. Due to its high hydrophobicity, BDQ might accumulate in lysosomal membranes and thereby change transmembrane ion permeability. In agreement with this hypothesis, it has been shown that BDQ can accumulate at the lipid membrane of liposomes and act as a H^+^/K^+^ ionophore (Hards et al., 2018). The ensuing lysosomal stress could then facilitate the dissociation of mTOR from the lysosomal membrane (Plescher et al., 2015) and the activation of TFEB. Interestingly, BDQ increases the expression of mucolipin 1 gene (*mcoln1*, Figure 3A and Figure 3—figure supplement 2A). Recently, Medina et al. showed that lysosomal Ca^2+^ is released through MCOLN1 and activates calcineurin, which binds and dephosphorylates TFEB (Medina et al., 2015). These two hypotheses are now mentioned in the Discussion section.

4) Although the synergistic effect of BDQ with PZA seems to approach 1 log (~10% survival) killing of M. tuberculosis, the effect size of BDQ alone (Figure 3F) is quite small. This raises questions of how significant these observations are to TB. This is important, since the whole setup is to find new therapeutic agents to control MDR TB.

In the former Figure 3F, we used a low concentration of BDQ (1 µg/mL) in order to discriminate between additive and synergic effect. At higher concentration of BDQ (5 µg/mL, the concentration detected in the plasma of TB patients treated with BDQ (Andries et al., 2005)), we observed a 50% decrease in the number viable bacteria (now in Figure 6F).

5) The authors use BAPTA (extracellular) to eliminate calcium and test the effect of loss of TFEB activation on bacterial CFU. This is done with *S. aureus* (not TB). Also, if they really are using BAPTA, they would not be inhibiting intracellular calcium release but import from extracellular stores, changing the interpretation of the experiment.

The calcium chelation experiments were performed with BAPTA-AM, not BAPTA. We would like to apologize for this error and have corrected it in the revised manuscript. Unlike BAPTA, which is a membrane-impermeable calcium chelator that binds extracellular calcium ions, BAPTA-AM is a cell permeable analog of BAPTA that binds calcium only after the acetoxymethyl group is removed by cytoplasmic esterases (Tymianski et al., 1994). It is highly selective for Ca^2+^ over Mg^2+^, and It is commonly used to evaluate the role of intracellular Ca^2+^ in cell signaling.

[Editors’ note: what follows is the authors’ response to the second round of review.]

Reviewer #1:[…]My only query relates to new data (Figure 7) showing that human macrophages induce nitrite, which the authors claim is derived from host NO. As far as I am aware, there has been considerable difficulty demonstrating induction of NO by human macrophages in culture. In some cases, the NO2- observed has actually been produced by the bacteria themselves (see Bussel, Zhang and Nathan, 2013). The authors may want to take this into account.

We thank the reviewer for his/her positive comments and suggestions, which have enabled us to improve our manuscript significantly. We agree the role of NO in the inhibition of *M. tuberculosis* growth is controversial in human. However, in Figure 7D, we quantified NO2- in uninfected cells upon BDQ treatment, which excludes nitric oxide produced by bacteria.